



# Extending water vapor measurement capability of photon limited differential absorption lidars through simultaneous denoising and inversion

Willem J. Marais[1] and Matthew Hayman[2]

[1]Space Science Engineering Center, University of Wisconsin-Madison, Madison, Wisconsin, USA
[2]National Center for Atmospheric Research, Boulder, Colorado, USA

**Correspondence:** Willem J. Marais (willem.marais@ssec.wisc.edu)

**Abstract.** The MicroPulse DIAL (MPD) was developed at Montana State University (MSU) and the National Center for Atmospheric Research (NCAR) to perform range resolved water vapor (WV) measurements using low power lasers and photon counting detectors. The MPD has proven to produce accurate WV measurements up to 6 km altitude. However, the MPD's ability to produce accurate higher altitude WV measurements is impeded by the current standard Differential Absorption Lidar

(DIAL) retrieval methods. These methods are built upon a fundamental methodology which algebraically solves for the WV using the MPD forward models and noisy observations, which exacerbate any random noise in the lidar observations.

The work in this paper introduces the adapted Poisson Total Variation (PTV) specifically for the MPD instrument. PTV was originally developed for a ground based high spectral resolution lidar, and this paper reports on the adaptations that were required in order to apply PTV on MPD WV observations. The adapted PTV method, coined PTV-MPD, extends the maximum

altitude of the MPD from 6 km to 8 km and substantially increases the accuracy of the WV retrievals starting above 2 km. PTV-MPD achieves the improvement by simultaneously denoising the MPD noisy observations and inferring the WV by separating the random noise from the non-random WV.

An analysis with 130 radiosonde (RS) comparisons shows that the relative root mean square difference (RRMSE) of WV measurements between RS and PTV-MPD exceeds 100% between 6 and 8 km, whereas the RRMSE between RS and the

standard method exceeds 100% near 3 km. In addition, we show that by employing PTV-MPD, the MPD is able to extend its useful range of WV estimates beyond that of the ARM Southern Great Planes Raman lidar (RRMSE exceeding 100% between 3 and 4 km); the Raman lidar has a power-aperture-product 500 times greater than that of the MPD.

## 1 Introduction

Water vapor (WV) is one of the fundamental thermodynamic variables that defines the state of the atmosphere and influences

many important processes related to weather and climate. The importance of continuously monitoring lower tropospheric WV is underscored in the National Aeronautics and Space Administratio (NASA) decadal survey (National Academies of Sciences and Medicine, 2018b) and in National Research Council (NRC, 2009, 2010, 2012) and National Academy of Sciences (National Academies of Sciences and Medicine, 2018a) reports. In particular, continuous range resolved measurements of WV are needed





at large scales to improve severe weather and precipitation predictions (Weckwerth et al., 1999; Wulfmeyer et al., 2015; Geerts
et al., 2016; Jensen et al., 2016).

To fulfill this observational need Montana State University (MSU) and the National Center for Atmospheric Research
(NCAR) have developed a MicroPulse Differential absorption lidar (MPD) that continuously measures range resolved WV
in the lower (150 m to 6 km) atmosphere (Spuler et al., 2015, 2021; NCAR/EOL Remote Sensing Facility). The MPD is
designed for unattended network deployment, using low power, low cost, high reliability diode lasers that enable class 1M eye-
safe transmitted lasers. The MPD employs the narrowband differential absorption lidar (DIAL) technique on a WV absorption
line with low temperature sensitivity whereby approximate knowledge of atmospheric temperature and pressure allow for a
first principles based retrieval (Nehrir et al., 2009). An immediate benefit then of the MPD is that it provides continuous WV
measurements independent of radiosonde WV measurements, which can provide additional information to Numerical Weather
Prediction (NWP) data assimilation systems.

## 1.1  Problem statement - extending capability of MPD

The MPD has proven to produce accurate WV measurements *up to* 6 km altitude, depending on aerosol loading, clouds and
solar background. However, in high solar background conditions, MPD water vapor retrievals can be noisy as low as 2 km. The
MPD's ability to produce precise measurements above 2 km and accurate higher altitude WV measurements is impeded by
the current standard WV DIAL retrieval methods. These methods are built upon a fundamental methodology that algebraically
solves the WV variable through operations that exacerbate any random noise in the lidar observations (Marais et al., 2016). To
suppress the noise the standard method applies low pass filters on the algebraically computed WV using a Gaussian smoothing
kernel or Savitzky–Golay filter (Schafer, 2011). The signal-to-noise ratio (SNR) of the WV measurements can be further
improved upon by reducing the vertical and horizontal resolutions of the photon counting observations. However, reducing
the resolutions introduces systematic biases in the WV due to the non-linearity of the single scatter lidar equation. Smoothing
operations are limited in their benefits because optimal averaging is generally localized to patches of correlated structure in
a lidar profile. Invariably, parts of the profile are oversmoothed (resulting in biases), while other parts are undersmoothed
(resulting in random error) (Hayman et al., 2020).

Fig. 1 illustrates the shortcomings of low pass filtering, where Fig. 1.b and Fig. 1.e show the standard method estimated WV
image and profile obtained using low pass filtering. The profile is juxtaposed with a coincident radiosonde (RS) WV profile.
Although the bandwidth of the low pass filter is relatively well suited for low altitude ($\leq$ 4 km) WV measurements, Fig. 1.i
shows that the filter is not sufficient to meaningfully reduce the noise at higher altitudes. Furthermore, Fig. 1.ii illustrate that
the low pass filter also over-smooths rapidly changing WV features such as embedded dry regions.

## 1.2  Proposed approach to extending MPD capability

Advances have been made in denoising photon counting medical images where the photon detection methodologies and for-
ward modeling are similar to that in atmospheric lidar (Fessler, 2020; Oh et al., 2013; Harmany et al., 2012; Willett and
Nowak, 2003; Ahn and Fessler, 2003). Basically, these methods quantitatively separate the image being estimated from the



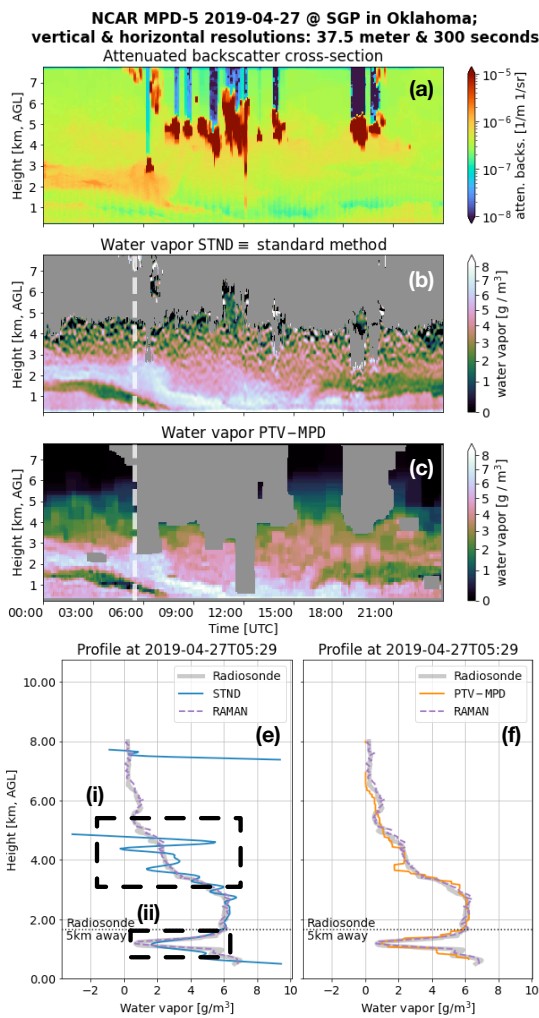

**Figure 1.** The (a) attenuated backscatter cross-section (atten. backs.), the water vapor (WV) measurements of the (b) standard (`STND`) and (c) PTV-MPD methods. (a) is not masked to show what atmospheric features have been been masked out in (b) and (c) by the gray areas. The white dashed vertical lines in (b) and (c) show the launch time of the radiosonde (RS), where (e) and (f) show the RS WV profile. (e) compares the standard method WV measurements with RS WV profile, whereas (f) shows the same comparison for the PTV-MPD method. The horizontal dashed lines in (e) to (f) show the altitude at which the RS was horizontally 5 km away from the MPD. The dashed boxes in (e) and (f) highlight regions that are discussed in the text.

random noise by making a distinction between the 1) vertical and horizontal correlations among the pixels of the underlying image and 2) the uncorrelated photon counting noise. The work of (Xiao et al., 2020; Hayman and Spuler, 2017; Marais et al., 2016) have demonstrated that the medical denoising methods can be adopted and adapted to dramatically improve the inference of the extinction cross-section from ground-based photon counting high spectral resolution lidars (HSRLs).





Inspired by the medical image denoising methods and the Poisson Total Variation (PTV) method for inferring extinction from HSRL observations (Xiao et al., 2020; Hayman and Spuler, 2017; Marais et al., 2016), we explore how the PTV method can be adapted for the MPD instrument to extend its capability measuring WV. Figures 1.c and 1.f show an example WV measurement of the adapted PTV method, labeled as PTV-MPD, of the same scene and profile of Figures 1.c and 1.e; using

the range intervals indicated by Figures 1.i and 1.ii the PTV-MPD WV measurements are more accurate at higher altitudes and at the embedded dry region compared to the standard method.

The PTV-MPD method we present in this paper differs from PTV by three adaptations that are necessary for accurate WV measurements (Marais et al., 2016).

*Forward models for DIAL*: The first adaptation of PTV requires that we develop MPD forward models to fit the estimated

parameters to the observed photon counts. While the objective is to estimate the WV, to accommodate the forward model, the attenuated backscatter (atten. backs.) is also an estimated product. In addition, the MPD instrument transmits laser pulses that are longer than the range resolution of the captured data. Hence, the forward model also includes a convolution operation with the modeled laser pulse that blurs the single scatter lidar equation (SSLE).

*Simultaneous inference*: PTV for the HSRL infers the backscatter and extinction cross-sections with a two-step approach

in order to isolate geometric overlap calibration biases to the extinction cross-section. However, with PTV-MPD a two-step approach is not be necessary to infer the WV and atten. backs. since the DIAL employs a differential measurement technique which decouples the WV measurement from the geometric overlap. Hence, the second adaptation of PTV is that PTV-MPD infers the WV and atten. backs. simultaneously instead of separately. The simultaneous inference approach produces WV estimates that are more faithful to the MPD observations compared to the two-step inference approach.

*Mitigate DIAL sensitivity to initial atten. backs.*: The simultaneous inference of the WV and atten. backs. requires initial estimates of both the WV and atten. backs. Our experiments show that PTV-MPD is sensitive to near range inaccuracies in the initial atten. backs. estimate which induce inaccuracies in the near range WV measurements. Therefore, the third adaptation involves making PTV-MPD more robust against inaccuracies in the initial atten. backs.

### 1.3   Contributions

The immediate contributions of this paper are the

1. adaptation of the PTV method (i.e. PTV-MPD) for the DIAL technique,

2. the first rigorous validation of the PTV method using in situ measurements.

This work also serves as an example how development of advanced signal processing can provide insights into improving hardware design and trades. By leveraging advances in signal processing, lidar hardware costs might be reduced whilst

maintaining or improving the retrieval precision.





### 1.4 Outline, notation and symbols

We introduce the MPD forward models in Sect. 2. Thereafter, in Sect. 3 we discuss the noise model of the MPD and how saturated photon counts are masked. The standard and PTV-MPD methods are discussed in Sect. 4 and Sect. 5. The paper ends with experiment results in Sect. 6 and thereafter the conclusion.

95        Geophysical variables and the lidar forward models are written as matrices. When we introduce a geophysical related variable we immediately indicate the units of the variable using the notation $[\cdot]$; for example [W]. Table 1 lists the primary matrices and index variables used throughout this paper. The $n$-th row and $k$-th column element of a matrix is denoted for example by $\varphi_{n,k}$. We use the superscript index $(\iota)$ to indicate whether a matrix or vector is specific to lidar channel (i.e. wavelength).

Table 2 lists commonly used acronyms that are being used throughout this paper.

## 2    The MPD forward models

To date NCAR has five experimental MPD instruments where each instrument transmits laser pulses at a rate of 7 to 8 kHz at two wavelengths sequentially. The first wavelength is tuned *on* a WV absorption line near 828.2 nm, whereas the second wavelength is *off* an WV absorption line near 828.3 nm (Spuler et al., 2015); these wavelengths are indexed by $\iota \in \{\mathrm{on}, \mathrm{off}\}$. The MPDs transmit laser pulses that are longer than the sampling interval $\Delta t = 250$ ns to increase the SNR of the WV measurements; the laser pulse duration was $1\mu$s during a Southern Great Plains (SGP) field campaign and has recently been decreased to 625 ns for the next generation MPDs (Spuler et al., 2021). The integer $\Delta N \geq 0$ quantifies how many vertical sampling intervals $\Delta t$, minus 1, fits in duration of a laser pulse minus one; for example, for the SGP data $\Delta N = (10^{-6}/\Delta t) - 1 = 3$.

The MPD instrument photon detector observes the weighted sum of the lidar equation and the laser pulse, since the laser pulse spans multiple range sampling intervals. We define the SSLE on the measurement range axis which are denoted by $\boldsymbol{r}_{n'}$ where $n' = 1, 2, \ldots, N + \Delta N$. The MPD samples the photon rates mid-duration of the laser pulse, and therefore the range axis of the observations is shifted relative to the measurement range axis. Specifically, the observation range axis is denoted by $\tilde{\boldsymbol{r}}_n$ where $n = 1, 2, \ldots, N$ indexes the observation range axis; the relation between the observation $\tilde{\boldsymbol{r}}_n$ and measurement $\boldsymbol{r}_{n'}$ range axes is

$$\tilde{\boldsymbol{r}}_n = \boldsymbol{r}_{n+\lceil \Delta N/2 \rceil}. \tag{1}$$

The SSLE is a function of the unknown WV $\varphi_{n',k}$ [g m$^{-3}$] and uncalibrated atten. backs. $\chi_{n',k}$ [m$^{-1}$ sr$^{-1}$] where the columns of the matrices are indexed by $k = 1, 2, \ldots K$. We write the SSLE as a matrix function since our attention is on inferring an image of the WV. The single scatter lidar matrix function at measurement range $\boldsymbol{r}_{n'}$ for channel $\iota$ is denoted by

$$\tilde{\mathbf{S}}_{n',k}^{(\iota)}(\boldsymbol{\varphi}, \boldsymbol{\chi}) = \frac{\mathbf{C}_{n',k}}{\Delta t\, \boldsymbol{r}_{n'}^2} \boldsymbol{O}_{n'} \boldsymbol{\chi}_{n',k} \tag{2}$$

$$\times \exp\left( -2\Delta r \sum_{n''=0}^{n'} \boldsymbol{\sigma}_{n'',k}^{(\iota)} \boldsymbol{\varphi}_{n'',k} \right) \tag{3}$$





and its unit is [W]. The calibration parameters are the range resolution $\Delta r = c\Delta t/2 \approx 37.5\text{m}$, backscatter calibration constant $\mathbf{C}_{n',k}$ [J sr m$^3$] and geometric overlap $\boldsymbol{O}_{n'}$. The WV absorption [m$^2$ g$^{-1}$] of channel $\iota$ is denoted by $\boldsymbol{\sigma}_{n',k}^{(\iota)}$ and is pre-computed using 1) an assumed lapse rate with which the temperature and pressure profiles are approximated, 2) the mass of a water molecule per mole and 3) the Avogadro number (Spuler et al., 2015; Nehrir et al., 2009).

125    The weighted sum of the SSLE with the laser pulse is modeled by

$$\Delta t \sum_{n'=n}^{n+\Delta N} \mathbf{A}_{n,n'} \tilde{\mathbf{S}}_{n',k}^{(\iota)}(\boldsymbol{\varphi},\boldsymbol{\chi}). \tag{4}$$

where the matrix $\mathbf{A}_{n,n'}$ models the laser energy distribution over the multiple range sampling intervals; the factor $\Delta t$ models the integration of the observed photon rate by the photon detectors. The first $\Delta N + 1$ columns ($n' = 1, 2, \ldots, \Delta N + 1$) of the first row ($n = 1$) of the matrix $\mathbf{A}_{n,n'}$ is the fractional laser energy distribution over the duration of the pulse such that the sum of the first row is equal to one, and the remaining $N$ columns are equal to zero. The $n$-th row of matrix $\mathbf{A}_{n,n'}$ is the first row of $\mathbf{A}_{n,n'}$ shifted circularly to the right $n$ times. Hence, the MPD forward model at range $\tilde{r}_n$ for channel $\iota$ is defined by

$$\mathbf{S}_{n,k}^{(\iota)}(\boldsymbol{\varphi},\boldsymbol{\chi}) \tag{5}$$
$$= \boldsymbol{U}_k \Delta t \left( \boldsymbol{b}_k^{(\iota)} + \sum_{n'=n}^{n+\Delta N} \mathbf{A}_{n,n'} \tilde{\mathbf{S}}_{n',k}^{(\iota)}(\boldsymbol{\varphi},\boldsymbol{\chi}) \right),$$

where $\boldsymbol{U}_k$ is the number of laser shots per column index $k$. The vector $\boldsymbol{b}_k^{(\iota)}$ is the dark and solar background photon rate [W] of channel $\iota$.

The MPD instrument photon detectors are saturated over several range bins after each laser shot due to internal scattering in the co-axial optical configuration. Consequently, the first WV estimate starts at 125 m or 500 m depending on the hardware configuration. To model the unobserved WV from range 0 up to $\boldsymbol{r}_1$, the lowest water absorption cross-section $\boldsymbol{\sigma}_{1,k}^{(\iota)}$ is the range axis sum of the absorption cross-sections from range 0 up to $\boldsymbol{r}_1$. Hence, the WV $\boldsymbol{\varphi}_{1,k}$ at range $\boldsymbol{r}_1$ represents the range-average WV from range 0 up to $\boldsymbol{r}_1$.

## 3  The MPD photon counting noise model and masking

### 3.1  Photon counting noise probability mass function

The photon counting observations at range $\tilde{r}_n$ and profile $k$ of the on- and offline channels are denoted by $\mathbf{Y}_{n,k}^{(\iota)}$. Each photon count represents temporally accumulated counts of multiple laser shots as indicated by $\boldsymbol{U}_k$. The noise of the photon counts are modeled by the Poisson probability mass function (PMF) if the corresponding photon rates are below the saturation limit of the MPD photon detectors (Donovan et al., 1993; Müller, 1973). The expected values of these unsaturated photon counts is





modeled by the MPD forward model Eq. (5) and we assume that

$$\mathbf{Y}_{n,k}^{(\iota)} \sim \text{Poisson}\left(\mathbf{S}_{n,k}^{(\iota)}\right) \tag{6}$$

$$\equiv \exp\left(-\mathbf{S}_{n,k}^{(\iota)} + \mathbf{Y}_{n,k}^{(\iota)} \log_e \mathbf{S}_{n,k}^{(\iota)} - \log_e \mathbf{Y}_{n,k}^{(\iota)}!\right)$$

where $\mathbf{S}_{n,k}^{(\iota)} \equiv \mathbf{S}_{n,k}^{(\iota)}(\boldsymbol{\varphi}, \boldsymbol{\chi})$. $\tag{7}$

## 3.2   Masking saturated photon counts

The instantaneous backscattered photon rates, corresponding to each laser shot, of clouds and precipitation can exceed the MPD photon detector saturation limit. Moreover, the accumulated photon counts $\mathbf{Y}_{n,k}^{(\iota)}$ consists of a combination of un- and saturated photon counts within and at the edges of clouds and precipitation. Photon counts that are saturation contaminated

cannot be accurately modeled by the Poisson PMF (Donovan et al., 1993). Hence, saturated photon counts and corresponding forward model pixels are excluded from our inference methodology using a mask matrix $\mathbf{M}_{n,k}$.

The saturated photon count mask $\mathbf{M}_{n,k}$ has to be constructed via proxy data that indicate whether the instantaneous backscattered photon rates exceed the photon detector saturation limit. Since we know a priori that dense aerosol layers, clouds and precipitation have large backscatter cross-sections, the matrix $\mathbf{M}_{n,k}$ masks out these atmospheric features.

We used a sliding window standard deviation filter on the photon counts $\mathbf{Y}_{n,k}^{(\iota)}$, with a fixed threshold, to identify the large backscatter cross-sections of dense aerosol layers, clouds and precipitation; the threshold was set by qualitatively validating whether these large backscatter cross-sections have been masked out.

## 3.3   Photon counting noise negative log likelihood

The Poisson negative log likelihood (P-NLL) is used by the PTV-MPD method to quantify the fitting of the WV and atten.

backs. onto the noisy observations via the MPD forward model Eq. (5). The P-NLL function of channel $\iota$ is denoted by

$$\mathcal{L}_p\left(\mathbf{S}^{(\iota)}(\boldsymbol{\varphi}, \boldsymbol{\chi}); \mathbf{Y}^{(\iota)}\right) = \sum_{n=1,k=1}^{N,K} \mathbf{M}_{n,k} \tag{8}$$

$$\times \left[p\mathbf{S}_{n,k}^{(\iota)}(\boldsymbol{\varphi}, \boldsymbol{\chi}) - \mathbf{Y}_{n,k}^{(\iota)} \log_e\left(p\mathbf{S}_{n,k}^{(\iota)}(\boldsymbol{\varphi}, \boldsymbol{\chi})\right)\right],$$

where $p$ is a normalization factor that is used in conjunction with the photon counts. The saturated photon counts are masked out with the matrix $\mathbf{M}_{n,k}$. The factorial term of the Poisson PMF is omitted in Eq. (8) since it is a constant value.

# 4   The MPD standard method

A detailed discussion of the standard processing technique employed for MPD is provided in (Spuler et al., 2021). Nonetheless, we will provide a brief overview of the method here as outlined in Alg. 1.

The photon count profiles first undergo low pass filtering to suppress the random noise (Hayman et al., 2020). The ratio of the on- and off-line channels cancels the atten. backs. in each forward model, which allows us to solve directly for the



---

**Algorithm 1** The MPD standard method.

---

**Require:** The noisy observations $\mathbf{Y}_{n,k}^{(\mathrm{on})}$ and $\mathbf{Y}_{n,k}^{(\mathrm{off})}$.

1: {Low pass filter background subtracted photon counts}

$$\tilde{\mathbf{Y}}_{n,k}^{(\iota)} = \mathrm{lowpass-filter}\left(\mathbf{Y}_{n,k}^{(\iota)} - \boldsymbol{U}_k \Delta t \boldsymbol{b}_k^{(\iota)}\right) \tag{9}$$

2: {Compute differential optical depth}

$$\boldsymbol{\tau}_{n,k} = -\frac{1}{2}\log_e\left(\frac{\tilde{\mathbf{Y}}_{n,k}^{(\mathrm{off})}}{\tilde{\mathbf{Y}}_{n,k}^{(\mathrm{on})}}\right)$$

3: {Compute the absolute water vapor}

$$\boldsymbol{\tau}_{0,k} = \mathbf{0}, \quad \tilde{\boldsymbol{\varphi}}_{n,k} = \frac{\boldsymbol{\tau}_{n,k} - \boldsymbol{\tau}_{n-1,k}}{\boldsymbol{\sigma}_{n'',k}^{(\mathrm{on})} - \boldsymbol{\sigma}_{n,k}^{(\mathrm{off})}}$$

4: {Low pass filter the computed water vapor}

$$\hat{\boldsymbol{\varphi}}_{n,k} = \mathrm{lowpass-filter}(\tilde{\boldsymbol{\varphi}}_{n,k})$$

5: **return** the water vapor estimate $\hat{\boldsymbol{\varphi}}_{n,k}$.

---

optical depth resulting from WV. At this stage, the WV estimate has residual noise that are low pass filtered with Gaussian kernels, with 5 to 10 minute by 170 m bandwidths, to further reduce the noise in the observation. The smoothing kernel of the low pass filter is fixed across the scene. Thus, the WV estimates are often oversmoothed in highly dynamic, high SNR regions and undersmoothed at higher altitudes. Due to the direct nature of the inversion, there are also no constraints imposed on the retrieval, therefore non-physical WV estimates are frequently obtained in noisy regions. The non-physical WV values,

in turn, can create problems in further downstream scientific analysis where non-physical state variables present a challenge. Nonphysical quantities are generally not easily included in such analysis but the selective omission or limiting of non-physical noise will also create biases.

Finally, the length of the laser pulses are not accounted for in the standard method. Specifically, inclusion of the laser pulse in the forward model prevents the atten. backs. term from directly canceling out when dividing the offline with the online channel

and a direct inversion becomes no longer possible. However, failing to account for the laser pulse length can create biases in some cases, for example at WV dry regions.





## 5 The PTV-MPD method

The PTV method, originally developed for photon counting HSRL (Marais et al., 2016), is an adaptation of the SPIRAL method which is a regularized maximum likelihood technique (Oh et al., 2013; Harmany et al., 2012); the technique and derivations of it have be applied in wide range of inverse problems in different domains such as in medical imaging and astronomy (Roelofs et al., 2020; Harmany et al., 2012).

### 5.1 Overview of PTV

With PTV the assumptions are that

1. the photon counting noise can be accurately modeled by the Poisson PMF in Eq. (6),

2. the expected value of the photon counts can be accurately modeled with a forward model (i.e. Eq. (5)) and

3. the geophysical variable (i.e. WV) image that we want to estimate can be accurately approximated with a two-dimensional (2D) piecewise constant (PC) function.

An accurate noise model is important in inverse problems since with Poisson noise the noise variance is signal dependent and modeling the photon counting noise as Gaussian distribution will lead to inversion inaccuracies (Harmany et al., 2012). The 2D PC approximation induces spatial correlations on the estimated geophysical variable while preserving any discontinuities; such correlations are expected from the true unobserved geophysical variable of which WV and atten. backs. are examples.

By making a distinction between the random noise and the geophysical variable having spatial correlation, PTV is capable of separating the noise from the geophysical variable that is being estimated. PTV achieves this separation by 1) using the P-NLL from Eq. (8) to quantify how close of a fit the forward model is relative to the noisy observations and by 2) employing the total variation (TV) penalty function that regularizes the geophysical variable to be approximately 2D PC.

The approach used by PTV is formulated as a mathematical optimization framework where we search over all candidate geophysical variables and choose the geophysical variable that minimizes the sum of the 1) P-NLL composited with the forward model and 2) the TV penalty functions. The P-NLL composite is called the *loss function* and the loss summed with the penalty functions is called the *objective function*. The optimization framework is conceptually illustrated by the equation

$$\min_{\boldsymbol{X}} \Big\{ \underbrace{\overbrace{\tilde{\ell}(\boldsymbol{X};\boldsymbol{Y})}^{} + \underbrace{\tilde{\lambda}}_{\text{Tuning parameter}} \times \underbrace{\text{TV}(\boldsymbol{X})}_{\text{TV penalty function}}}_{\text{Objective function}} \Big\} \tag{10}$$

where

$\boldsymbol{X} \equiv$ geophysical variable,

$\boldsymbol{Y} \equiv$ photon counting observations

$\tilde{\ell}(\boldsymbol{X};\boldsymbol{Y}) \equiv$ P-NLL composited with forward

model, e.g. Eq. (8).


The (anisotropic) TV penalty function is defined as (Harmany et al., 2012)

$$
\mathrm{TV}(\boldsymbol{X}) \qquad \qquad (11)
$$

$$
= \sum_{\substack{n=1 \\ k=1}}^{\substack{N-1 \\ K}} |\boldsymbol{X}_{n,k} - \boldsymbol{X}_{n+1,k}| + \sum_{\substack{n=1 \\ k=1}}^{\substack{N \\ K-1}} |\boldsymbol{X}_{n,k} - \boldsymbol{X}_{n,k+1}|,
$$

and is weighted with a tuning parameter $\tilde{\lambda} \geq 0$ that sets the degree to which the 2D PC regularization is promoted. The tuning

parameter is quantitatively determined through a cross-validation methodology (Friedman et al., 2001, chap. 7)(Oh et al., 2013), and the parameter is not set by "expert opinion".

## 5.2   PTV-MPD: Adaptation required for MPD

We adopt the PTV method to infer the WV from the MPD observations where we approximate both the WV and atten. backs. as 2D PC functions. In contrast, the standard method Alg. 1 divides out the atten. backs., which is not feasible with PTV-MPD

since it is unclear how to model the noise of the ratio of background subtracted photon counts.

In order to apply PTV on MPD observations the following adaptations are required:

**(A1)**  Use the MPD forward models Eq. (5),

**(A2)**  allow the simultaneous inference of the WV and atten. backs.,

**(A3)**  and prevent the degradation of the WV measurement due to inaccuracies in initial the atten. backs. value.

Adaptations **(A1)** and **(A2)** are achieved by redefining the PTV loss and objective functions as discussed in the next section. Specific to adaptation **(A2)** in the original HSRL formulation of PTV 1) the backscatter cross-section was computed from the denoised photon counts and 2) the lidar ratio was inferred using the computed backscatter cross-section. This two-step sequence isolate calibration biases due to the geometric overlap function from the inferred lidar ratio and the extinction cross-section. With a DIAL instrument such as MPD, the geometric overlap does not induce a bias in the inferred WV in part because

both measurements use the same photon detector (Spuler et al., 2015, 2021). Therefore, the two-step sequence used for HSRL observations is not necessary.

Regarding adaptation **(A3)**, the simultaneous inference of the WV and atten. backs. requires an initial value of the atten. backs. During our investigation of adapting PTV for the MPD instrument we noticed that the adapted PTV method is sensitive to inaccuracies in the initial atten. backs. value. Specifically, when computing the initial value of the atten. backs. algebraically

from the photon counting observations, the contributions to the inaccuracies in the initial atten. backs. originate from the small SNR at close ranges and the long laser pulses that convolves with the SSLE (see Eq. (4)). Hence, for adaptation **(A3)** we propose that PTV-MPD to employ a coarse-to-fine image resolution framework when inferring the WV (Marais and Willett, 2017). With the coarse-to-fine framework we first infer the WV and atten. backs. at a coarse image resolution which allows for reducing inaccuracies in the initial atten. backs. In other words, at the coarse image resolution the long laser pulses are





deconvolved from the initial atten. backs. and the SNR of close range observations are implicitly increased. In a sequence from coarse-to-fine image resolution we use the coarse resolution WV and atten. backs. estimates as an initial values when inferring the finer image resolution WV and atten. backs.

### 5.3 PTV-MPD method algorithmic details

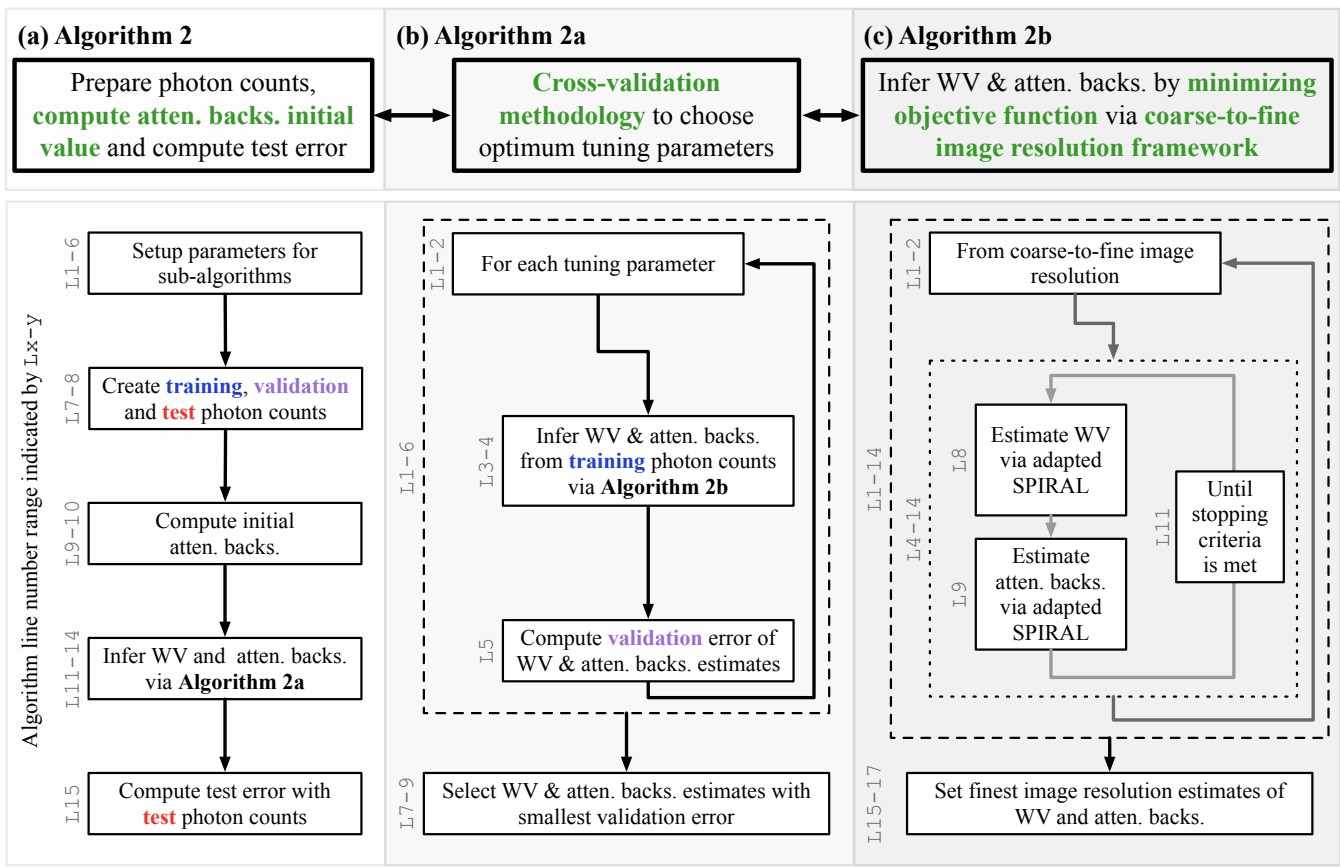

**Figure 2.** Pictorial overview of Algorithm 2 and its sub-algorithms Alg. 2a and Alg. 2b for inferring the water vapor (WV) and uncalibrated attenuated backscatter cross-section (atten. backs.). The corresponding algorithm line numbers for each block in the flow diagrams are indicated by the vertical text Lx-y. The blue, violet and red text are meant to highlight statistically independent datasets generated using Poisson thinning.

Using the conceptual optimization framework in Eq. (10) as a reference, the PTV-MPD method is formulated as an op-
timization problem where the minimizer of the objective function are the estimates of the WV $\varphi$ and atten. backs. $\chi$. The objective function used by PTV-MPD is introduced in Sect. 5.3.1 which addresses adaptations **(A1)** and **(A2)**; Sect. 5.3.1 serves as a preliminary section before elaborating on the coarse-to-fine image resolution framework. Minimizing an objective function requires initial values of the WV $\varphi$ and atten. backs. $\chi$ and can have a significant impact on the estimate of these





variables (Harmany et al., 2012; Kelley, 1999); to this effect in Sect. 5.3.2 we discuss the initial values of the WV $\varphi$ and atten.
backs. $\chi$. Finally, Sect. 5.3.3 discusses how the objective function is minimized through the coarse-to-fine image resolution
framework which addresses adaption **(A3)**.

The formulation of the PTV-MPD includes tuning parameters which are computed using a cross-validation (CV) method-
ology; the CV involves calculating a validation error that indicate the optimality of the tuning parameters and is discussed in
Sect. 5.3.4. Related to the tuning parameters and validation error, we require a way to test the hypothesis that the coarse-to-fine
image resolution framework will improve the accuracy of inferring the WV $\varphi$. To test the hypothesis, we compare the test
errors from inferring the WV $\varphi$ at only the finest image resolution versus from a coarse-to-fine image resolution. The test error
computation is also discussed in Sect. 5.3.4.

Fig. 2 gives a broad pictorial overview of how each part of the PTV-MPD framework is implemented with modularized
sub-algorithms; the purpose of the sub-algorithms are:

**Fig. 2.a, Alg. 2**  performs the necessary preparations to infer the WV $\varphi$ and atten. backs. $\chi$ which include computing the initial
atten. backs.

**Fig. 2.b, Alg. 2a**  employs a CV methodology to choose the optimum tuning parameters for inferring the WV $\varphi$ and atten.
backs. $\chi$.

**Fig. 2.c, Alg. 2b**  infers the WV $\varphi$ and atten. backs. $\chi$. for a fixed tuning parameter via coarse-to-fine image resolution frame-
work.

These algorithms are outlined in Sect. 5.3.5 which include specific details about the SPIRAL method adaptations.

### 5.3.1  Preliminary for coarse-to-fine framework: the loss and objective functions

The PTV-MPD loss function differs from the PTV loss function in two respects; cf. Eq. (23) in Marais et al. (2016). *First*,
the loss function of PTV-MPD is defined such that we infer the $\log_e$ of the atten. backs. denoted by $\tilde{\chi}_{n,k} \equiv \log_e \chi_{n,k}$, since
previous work suggests that more accurate inference can be achieved by inferring the $\log_e$ of a linear variable with methods
similar to PTV (Oh et al., 2013, 2014). *Second*, we define the PTV-MPD loss function as the sum of the normalized P-NLL
function of each channel, where the normalization ensures that each P-NLL function is equally weighted; this weighting is
done since the accumulated photon counts of the offline channels are generally larger than that of the online channel. Hence,
the PTV-MPD loss function is defined by

$$\ell_p\left(\varphi, \tilde{\chi}; \mathbf{Y}^{(\mathrm{on})}, \mathbf{Y}^{(\mathrm{off})}\right) \tag{12}$$

$$= \sum_{\iota \in \{\mathrm{on,off}\}} \underbrace{\mathcal{L}_p\left(\mathbf{S}^{(\iota)}\left(\varphi, \exp(\tilde{\chi})\right); \mathbf{Y}^{(\iota)}\right)}_{\text{channel } \iota \text{ P-NLL Eq. (8)}} \times \overbrace{\omega^{(\iota)}}^{\text{channel } \iota \text{ weight}}$$

which is the weighted sum of the P-NLL of each channel where each P-NLL is normalized by the root sum square (i.e.
Frobenius norm) of the photon counts that have not been masked out. The Frobenius norm of the mask photon counts is




defined by

$$\omega^{(\iota)} = \left\| \mathbf{M} \mathbf{Y}^{(\iota)} \right\|_F^{-1} = \left[ \sum_{\substack{n=1, \\ k=1}}^{N,K} \left( \mathbf{M}_{n,k} \mathbf{Y}_{n,k}^{(\iota)} \right)^2 \right]^{-1/2}. \tag{13}$$

The objective function that is minimized by PTV-MPD is defined as

$$\tilde{F}_{p,\lambda_w,\lambda_a} \left( \boldsymbol{\varphi}, \tilde{\boldsymbol{\chi}}; \mathbf{Y}^{(\mathrm{on})}, \mathbf{Y}^{(\mathrm{off})} \right) \tag{14}$$

$$\equiv \underbrace{\ell_p \left( \boldsymbol{\varphi}, \tilde{\boldsymbol{\chi}}; \mathbf{Y}^{(\mathrm{on})}, \mathbf{Y}^{(\mathrm{off})} \right)}_{\text{Loss function, Eq. (12)}} + \overbrace{\lambda_w \mathrm{TV}(\boldsymbol{\varphi})}^{\text{Regularize WV}} + \underbrace{\lambda_a \mathrm{TV}(\tilde{\boldsymbol{\chi}})}_{\text{Regularize atten. backs.}},$$

which is parameterized with the WV $\boldsymbol{\varphi}$ and atten. backs. $\tilde{\boldsymbol{\chi}}$ and their respective tuning parameters $\lambda_w$ and $\lambda_a$. The WV $\lambda_w$ and atten. backs. $\lambda_a$ tuning parameters set the degree to which these parameters should be regularized to be 2D PC. While minimizing the objective function (14), the WV $\boldsymbol{\varphi}$ is constrained to be non-negative which is denoted by the set $\mathcal{W}$. The objective function (14) is minimized using the alternating minimization method in conjunction with adaptations of SPIRAL (Beck and Tetruashvili, 2013; Harmany et al., 2012); the alternating minimization is elaborated in Sect. 5.3.5 and outlined by Alg. 2b.

### 5.3.2 Initial values of objective function

In Appendix B we show that for low SNR regions of the photon counting images there can be multiple estimates of the WV $\boldsymbol{\varphi}$ or atten. backs. $\tilde{\boldsymbol{\chi}}$ that minimizes the objective function (14). The dark and solar background photon rates $\boldsymbol{b}^{(\iota)}$ are the primary factors that determine the domains over which the objective function has a unique minimizer. Consequently, we expect that PTV-MPD will require accurate initial values of the WV $\boldsymbol{\varphi}$ and atten. back. $\tilde{\boldsymbol{\chi}}$ whenever the photon counts SNR are dominated by the background rate $\boldsymbol{b}^{(\iota)}$ such as at close range and high altitude regions.

It will be advantageous for the MPD to make WV measurements that are statistically independent of RS for the purpose of providing additional statistically independent information for NWP data assimilation. Thus, in this paper we set the initial WV to zero [gm$^{-3}$]; this WV initial value reflects that we assume no a priori information about the WV other than 1) the assumed lapse rate[1] of the atmosphere and 2) the assumption that the WV can be accurately approximated with 2D PC functions. This initial WV value is denoted by $\hat{\boldsymbol{\varphi}}^{\mathrm{init}}$.

The offline channel is less sensitive to WV absorption compared to the online channel. Hence, the initial value of the atten. backs., denoted by $\hat{\tilde{\boldsymbol{\chi}}}^{\mathrm{init}}$, is computed from the offline channel with the initial WV $\hat{\boldsymbol{\varphi}}^{\mathrm{init}}$ via

$$\hat{\tilde{\boldsymbol{\chi}}}_{n',k}^{\mathrm{init}} = \log_e \left( \frac{\sum_{n=1}^{N} \mathbf{A}_{n',n}^T \left( \mathbf{Y}_{n,k}^{(\mathrm{off})} - \boldsymbol{U}_k \Delta t \boldsymbol{b}_k^{(\iota)} \right)}{\boldsymbol{U}_k \Delta t \tilde{\mathbf{S}}_{n',k}^{(\mathrm{off})}(\hat{\boldsymbol{\varphi}}^{\mathrm{init}}, \mathbf{1})} \right). \tag{15}$$

The matrix $\mathbf{A}^T$ maps the observations to the measurement space; the matrix is the adjoint (i.e. transpose) of the laser energy distribution matrix $\mathbf{A}$. The vector $\boldsymbol{U}$ is the number of accumulated laser shots and $\Delta t$ models the integration of the observed photon rate by the photon detectors (see Sect. 2).

---
[1]From the lapse rate the WV absorption cross-section is computed; see Sect. 2.





### 5.3.3 The coarse-to-fine inference framework

With the coarse-to-fine image resolution inference framework we start with estimating a coarse resolution WV image, and use the coarse resolution estimate as an initial WV value at a finer image resolution; this technique has proven useful to more accurately denoising and invert images for low SNR photon limited application (Marais and Willett, 2017; Azzari and Foi,
2017). The coarse image resolution estimation is akin to downsampling the noisy observations to increase the SNR, though the difference is that with the proposed methodology the downsampling in the coarse-to-fine framework takes in account the MPD forward model (5).

Following the approaches delineated in (Marais and Willett, 2017; Azzari and Foi, 2017), we define:

1. A series of values $\bar{h} = h_1 > h_2 > \ldots > h_L = 1$ that represents the downsampling factors of the coarse-to-fine image
resolutions at which the WV will be inferred; $\bar{h}$ represents the coarsest image resolution.

2. The downsampling operator is denoted by $\mathcal{D}_h^{\downarrow}$ and its corresponding upsampling operator $\mathcal{D}_h^{\uparrow}$.

The downsampled image $\mathcal{D}_h^{\downarrow}\varphi$ will have approximately $N/h$ rows and $K/h$ columns, depending on how the downsample operator handles boundary pixels. The coarse resolution WV image is denoted by $\phi_h$, and its relationship with the finest image resolution WV image $\varphi$ is governed by the upsampling operator $\mathcal{D}_h^{\uparrow}$ where $\varphi = \mathcal{D}_h^{\uparrow}\phi_h$. The corresponding atten. backs. is
denoted by $\psi_h$ and is not downsampled like the WV; the subscript $h$ in $\psi_h$ indicates that it is paired with the downsampled WV $\phi_h$ estimate.

Let $l = 1, 2, \ldots, L$ represent the coarse-to-fine image resolution index. The coarse-to-fine repeats the following steps for each $l$ until the finest image resolution WV estimate is obtained.

**Step 1:** Define the current image resolution WV estimate $\phi_{h_l}$ and its paired atten. backs. estimate $\psi_{h_l}$. Also define the
current image resolution objective function

$$F_{p,\lambda_w,\lambda_a}\left(\mathcal{D}_{h_l}^{\uparrow}\phi_{h_l}, \psi_{h_l}; \mathbf{Y}^{(\mathrm{on})}, \mathbf{Y}^{(\mathrm{off})}\right) \equiv \ell_p\left(\mathcal{D}_{h_l}^{\uparrow}\phi_{h_l},\right. \tag{16}$$

$$\left.\psi_{h_l}; \mathbf{Y}^{(\mathrm{on})}, \mathbf{Y}^{(\mathrm{off})}\right) + \lambda_w \mathrm{TV}\left(\phi_{h_l}\right) + \lambda_a \mathrm{TV}\left(\psi_{h_l}\right).$$

**Step 2:** If $l = 1$ use the provided initial value of the WV $\phi_{h_l}$ and atten. backs. $\psi_{h_l}$ when minimizing the objective function (16). Otherwise, use the previous image resolution estimates as initial values for the current image resolution. Specifically, the
initial value for the WV is $\mathcal{D}_{h_l}^{\downarrow}\mathcal{D}_{h_{l-1}}^{\uparrow}\hat{\phi}_{h_{l-1}}$ and for the atten. backs. it is $\hat{\psi}_{h_{l-1}}$.

**Step 3:** Estimate the current image resolution WV $\phi_{h_l}$ and its paired atten. backs. $\psi_{h_l}$ by minimizing the objective function (16)

$$\hat{\phi}_{h_l}, \hat{\psi}_{h_l} \tag{17}$$

$$= \arg \min_{\phi_{h_l} \in \mathcal{W}, \psi_{h_l}} F_{p,\lambda_w,\lambda_a}(\overbrace{\mathcal{D}_{h_l}^{\uparrow}\phi_{h_l}}^{\text{Fitting is performed on native resolution...}}, \psi_{h_l}; \mathbf{Y}^{(\mathrm{on})}, \mathbf{Y}^{(\mathrm{off})})$$

...by upsampling coarse resolution WV image.





where $\mathcal{W}$ is the set of non-negative numbers.

**Final step:** If $l = L$, the final WV and atten. backs. estimates are denoted by $\hat{\varphi}_{\bar{h},\lambda_w,\lambda_a}$ and $\hat{\tilde{\chi}}_{\bar{h},\lambda_w,\lambda_a}$ meaning

$$\hat{\varphi}_{\bar{h},\lambda_w,\lambda_a} = \hat{\phi}_{h_L} \text{ and } \hat{\tilde{\chi}}_{\bar{h},\lambda_w,\lambda_a} = \hat{\psi}_{h_L}. \tag{18}$$

The subscripts $\bar{h}$, $\lambda_w$ and $\lambda_a$ indicate that the estimates of the WV $\varphi$ and atten. backs. $\tilde{\chi}$ are the specific to *coarse-to-fine image resolution configuration* parameterized by $\bar{h}$ and the tuning parameters $\lambda_w$ and $\lambda_a$.

In practice the minimization of the objective function (16) is done by alternating minimization between the WV and atten. backs. as indicated in Fig. 2.c and outlined by Alg. 2b.

### 5.3.4   Computing validation error to choose optimum tuning parameters, and computing test error

The validation and test errors are computed by basically comparing the WV $\varphi$ and atten. backs. $\tilde{\chi}$ estimates against noisy observations that are statistically independent of these estimates. This is achieved by thinning the non-masked off- $\mathbf{Y}^{(\text{off})}$ and
on-line $\mathbf{Y}^{(\text{on})}$ photon counts through the Poisson thinning technique (Marais et al., 2016; Oh et al., 2013; Hayman et al., 2020); these thinned photon counts are statistically independent of each other and are Poisson distributed (Cinlar, 2013, Chap. 4). Specifically, for each non-masked pixel in a photon counting image $\mathbf{Y}^{(\iota)}$, the Poisson thinning technique randomly samples the individual photon counts and the sampled photon counts are placed in three photon counting images. We call these three images the training $\mathbf{Y}^{(\iota)}_{\text{trn}}$, validation $\mathbf{Y}^{(\iota)}_{\text{vld}}$ and test $\mathbf{Y}^{(\iota)}_{\text{tst}}$ photon counting images. The expected value of the training, validation
and test photon counts relative to the original photon counts are expressed by the scalars $p_{\text{trn}} + p_{\text{vld}} + p_{\text{tst}} = 1$ such that $\mathbb{E}[\mathbf{Y}^{(\iota)}_{\text{trn}}] = p_{\text{trn}}\mathbb{E}[\mathbf{Y}^{(\iota)}]$ where $\mathbb{E}[\mathbf{Y}^{(\iota)}_{\text{vld}}]$ and $\mathbb{E}[\mathbf{Y}^{(\iota)}_{\text{tst}}]$ are similarly defined.

**Training photon counts**

The training photon counts $\mathbf{Y}^{(\iota)}_{\text{trn}}$ are used to infer the WV $\hat{\varphi}_{\bar{h},\lambda_w,\lambda_a}$ and atten. backs. $\hat{\tilde{\chi}}_{\bar{h},\lambda_w,\lambda_a}$ for specific WV $\lambda_w$ and atten. backs. $\lambda_a$ tuning parameters and coarse-to-fine configuration $\bar{h}$. Specifically, when using the training photon counts the
objective function in (16) will be equal to

$$F_{p_{\text{trn}},\lambda_w,\lambda_a}\left(\mathcal{D}^{\uparrow}_{h_l}\phi_{h_l}, \psi_{h_l}; \mathbf{Y}^{(\text{on})}_{\text{trn}}, \mathbf{Y}^{(\text{off})}_{\text{trn}}\right) \tag{19}$$

where $p$ in (16) has been replaced with $p_{\text{trn}}$.

**Validation photon counts**

The validation error of the estimates per tuning parameter is computed with the P-NLL (8) and the forward model (5) and is
denoted by

$$\text{err}_{\text{vld},\lambda_w,\lambda_a} \tag{20}$$
$$= \mathcal{L}_{p_{\text{vld}}}\left(\mathbf{S}^{(\text{on})}\left(\hat{\varphi}_{\bar{h},\lambda_w,\lambda_a}, \exp(\hat{\tilde{\chi}}_{\bar{h},\lambda_w,\lambda_a})\right); \mathbf{Y}^{(\text{on})}_{\text{vld}}\right)$$
$$+ \mathcal{L}_{p_{\text{vld}}}\left(\mathbf{S}^{(\text{off})}\left(\hat{\varphi}_{\bar{h},\lambda_w,\lambda_a}, \exp(\hat{\tilde{\chi}}_{\bar{h},\lambda_w,\lambda_a})\right); \mathbf{Y}^{(\text{off})}_{\text{vld}}\right).$$





The optimum tuning parameters corresponds to the smallest validation error and this give us the WV and atten. backs. estimates


$$\hat{\boldsymbol{\varphi}}_{\bar{h}}, \hat{\tilde{\boldsymbol{\chi}}}_{\bar{h}} = \arg \min_{\lambda_w, \lambda_a} \mathrm{err}_{\mathrm{vld}, \lambda_w, \lambda_a}. \tag{21}$$

**Test photon counts**

The test error for the estimates $\hat{\boldsymbol{\varphi}}_{\bar{h}}$ and $\hat{\tilde{\boldsymbol{\chi}}}_{\bar{h}}$ are also computed with the P-NLL (8) and the forward model (5) and is denoted by

$$\mathrm{err}_{\mathrm{tst}, \bar{h}} = \mathcal{L}_{p_{\mathrm{tst}}} \left( \mathbf{S}^{(\mathrm{on})} \left( \hat{\boldsymbol{\varphi}}_{\bar{h}}, \exp(\hat{\tilde{\boldsymbol{\chi}}}_{\bar{h}}) \right); \mathbf{Y}_{\mathrm{tst}}^{(\mathrm{on})} \right) \tag{22}$$


$$+ \mathcal{L}_{p_{\mathrm{tst}}} \left( \mathbf{S}^{(\mathrm{off})} \left( \hat{\boldsymbol{\varphi}}_{\bar{h}}, \exp(\hat{\tilde{\boldsymbol{\chi}}}_{\bar{h}}) \right); \mathbf{Y}_{\mathrm{tst}}^{(\mathrm{off})} \right).$$

To test the hypotheses whether the coarse-to-fine image resolution framework does improve the WV measurements, we evaluate the inequality

$$\mathrm{err}_{\mathrm{tst}, \bar{h}=9} < \mathrm{err}_{\mathrm{tst}, \bar{h}=1}. \tag{23}$$

### 5.3.5 The PTV-MPD algorithm

Alg. 2 and its sub-algorithms 2a and 2b outlines the PTV-MPD algorithms as described by Fig. 2. Alg. 2 takes as input the photon counts and the coarse image resolution at which PTV-MPD will start inferring the WV; the starting image resolution is expressed as a downsampling factor $\bar{h} \geq 1$. Alg. 2a outlines the CV methodology, and Alg. 2b outlines the coarse-to-fine WV inference image resolution framework.

The alternating minimization of the objective function (17) (Beck and Tetruashvili, 2013), employed by Alg. 2b, is achieved
by adaptations of SPIRAL where each adaptation is specific to the WV and atten. backs. variable (Oh et al., 2013; Harmany et al., 2012). The adaptation of SPIRAL includes replacing the original objective function and gradient matrix of the loss function; Appendix A presents the loss function gradient matrix along with the Jacobian matrix of the MPD forward model. The stopping criteria that is employed to stop the alternating minimization iterations in Alg. 2b uses the relative distance between consecutive estimates at the specific coarse image resolution. In more detail, if between iterations $t+1$ and $t$ the
relative distance

$$\frac{1}{2} \left( \frac{\left\| \boldsymbol{\phi}^{(t+1)} - \boldsymbol{\phi}^{(t)} \right\|_F}{\left\| \boldsymbol{\phi}^{(t+1)} \right\|_F} + \frac{\left\| \boldsymbol{\psi}^{(t+1)} - \boldsymbol{\psi}^{(t)} \right\|_F}{\left\| \boldsymbol{\psi}^{(t+1)} \right\|_F} \right) \tag{24}$$

is less than $10^{-5}$ then the loop in Alg. 2b terminates, where

$$\| \boldsymbol{\phi} \|_F = \left( \sum_{n=1, k=1}^{N, K} \phi_{n,k}^2 \right)^{1/2}. \tag{25}$$





---

**Algorithm 2** The PTV-MPD method.

---

**Require:** 1) The noisy observations $\mathbf{Y}^{(\text{on})}$ and $\mathbf{Y}^{(\text{off})}$, 2) (*mandatory*) the coarsest resolution downsampling factor $\bar{h}$, 3) (*optional*) initial
values of the WV $\hat{\boldsymbol{\varphi}}^{\text{init}}$, 4) (*optional*) tuning parameters for WV $\lambda_w$ and atten. backs. $\lambda_a$.

1: **if** $\lambda_a$ and $\lambda_w$ are not provided **then**

2:     $\Lambda_{\tilde{\chi}} = \Lambda_{\tilde{\chi}} = \left\{ 10^{-2+i/11} \right\}_{i=0}^{11} = \left\{ 10^{-2}, 10^{-1.91}, \dots, 10^2 \right\}$

3: **else**

4:     $\Lambda_{\tilde{\chi}} = \{\lambda_a\}$ and $\Lambda_{\tilde{\chi}} = \{\lambda_w\}$.

5: **end if**

6: If WV $\hat{\boldsymbol{\varphi}}^{\text{init}}$ is not provided, set $\hat{\boldsymbol{\varphi}}^{\text{init}} = \mathbf{0}$

7: {Split photon counts in training, validation and test counts}

8: $\mathbf{Y}_{\text{trn}}^{(\iota)}, \mathbf{Y}_{\text{vld}}^{(\iota)}, \mathbf{Y}_{\text{tst}}^{(\iota)} = \text{Poisson Thinning}\left( \mathbf{Y}^{(\iota)} \right)$

9: {Compute the atten. backs. initial value}

10: $\hat{\tilde{\boldsymbol{\chi}}}^{\text{init}} = $ plug initial WV $\hat{\boldsymbol{\varphi}}^{\text{init}}$ into Eq. (15) using scaled training photon counts $\mathbf{Y}_{\text{trn}}^{(\text{off})}/p_{\text{trn}}$

11: {Infer the WV and atten. backs.}

12:

$$\hat{\boldsymbol{\varphi}}_{\bar{h}}, \hat{\tilde{\boldsymbol{\chi}}}_{\bar{h}} = \text{Algorithm 2a}\bigg( \hat{\boldsymbol{\varphi}}^{\text{init}}, \hat{\tilde{\boldsymbol{\chi}}}^{\text{init}}, \bar{h}, \Lambda_{\tilde{\chi}}, \Lambda_{\tilde{\chi}}, \tag{26}$$
$$\mathbf{Y}_{\text{trn}}^{(\text{on})}, \mathbf{Y}_{\text{vld}}^{(\text{on})}, \mathbf{Y}_{\text{trn}}^{(\text{off})}, \mathbf{Y}_{\text{vld}}^{(\text{off})} \bigg)$$

13: {Compute the atten. backs.}

14: $\hat{\boldsymbol{\chi}}_{\bar{h}} = \exp\left( \hat{\tilde{\boldsymbol{\chi}}}_{\bar{h}} \right)$

15: Compute test error $\text{err}_{\text{tst},\bar{h}}$ via Eq. (22)

16: **return** $\hat{\boldsymbol{\varphi}}_{\bar{h}}, \hat{\boldsymbol{\chi}}_{\bar{h}}, \text{err}_{\text{tst},\bar{h}}$

---

## 395    6    Experiment results

In this section we quantify the accuracy of the PTV-MPD WV measurements juxtaposed with the standard method; we use the
mnemonics `STND` and `PTV` to make a distinction between the standard and PTV methods. We use RS WV measurements as
an *independent* reference to estimate WV accuracies. The MPD with coincident RS observations was stationed 1) in proximity
of the Atmospheric Radiation Measurement (ARM) SGP atmospheric observatory during Spring and Summer of 2019[2] and
2) and at the NCAR Marshall field site in Boulder Colorado during Winter and Spring of 2020 and 2021 (Spuler et al., 2021).
During the SGP MPD deployment the ARM Raman lidar was making continuous WV measurements; Table 3 compares the
laser power and telescope diameters of the Raman lidar and MPD.

---

[2]Information about the project is available at https://www.arm.gov/research/campaigns/sgp2019mpddemo.





---

**Algorithm 2a** Cross-validation methodology in choosing optimum tuning parameters when inferring WV $\varphi$ and atten. backs. $\chi$.

---

**Require:** 1) WV $\hat{\varphi}^{\text{init}}$ and atten. backs. $\hat{\tilde{\chi}}^{\text{init}}$ initial values, 2) the coarsest resolution downsampling factor $\bar{h}$, 3) tuning parameter sets $\Lambda_{\tilde{\chi}}$ and $\Lambda_{\tilde{\chi}}$, 4) training $\mathbf{Y}_{\text{trn}}^{(\text{on})}, \mathbf{Y}_{\text{trn}}^{(\text{off})}$ and validation $\mathbf{Y}_{\text{vld}}^{(\text{on})}, \mathbf{Y}_{\text{vld}}^{(\text{off})}$ photon counts.

  1: {Choose tuning parameters via cross-validation}

  2: **for** $(\lambda_w, \lambda_a) \in \Lambda_{\tilde{\chi}} \times \Lambda_{\tilde{\chi}}$ **do**

  3:     {Infer WV and atten. backs. using redefined objective function}

  4:     $\hat{\varphi}_{\bar{h}, \lambda_w, \lambda_a}, \hat{\tilde{\chi}}_{\bar{h}, \lambda_w, \lambda_a} =$
          Algorithm 2b$\left( \hat{\varphi}^{\text{init}} \hat{\tilde{\chi}}^{\text{init}}, \lambda_w, \lambda_a, \mathbf{Y}_{\text{trn}}^{(\text{on})} \mathbf{Y}_{\text{trn}}^{(\text{off})}, \bar{h} \right)$

  5:     Compute the validation error $\text{err}_{\text{vld}, \lambda_w, \lambda_a}$ via Eq. (20)

  6: **end for**

  7: {Choose tuning parameter that has smallest validation error}

$$\lambda_w^*, \lambda_a^* = \arg \min_{\lambda_w, \lambda_a} \{ \text{err}_{\text{vld}, \lambda_w, \lambda_a} \} \tag{27}$$

  8: {With chosen tuning parameters select estimates}

  9: $\hat{\varphi}_{\bar{h}} = \hat{\varphi}_{\bar{h}, \lambda_w^*, \lambda_a^*}, \hat{\tilde{\chi}}_{\bar{h}} = \hat{\tilde{\chi}}_{\bar{h}, \lambda_w^*, \lambda_a^*}$

10: **return** $\hat{\varphi}_{\bar{h}}, \hat{\tilde{\chi}}_{\bar{h}}$

---

## 6.1 Radiosonde comparison methodology

Although the MPD instrument was in close proximity of the ARM site during the SGP deployment the MPD, Raman lidar and
RS instruments all observe different atmospheric volumes at different altitudes especially in highly convective conditions. Furthermore, the Raman lidar uses some RS observations for calibration, and therefore the instrument is not entirely independent of the RS (Newsom and Sivaraman, 2018). Therefore, we

1. indicate the altitude at which the RS is horizontally more than 5 km away from the MPD,

2. and also show the ARM Raman lidar WV retrievals during the SGP deployment.

In contrast to the Raman lidar, the MPD retrievals or calibrations do not employ RS observations. Instead, the MPD retrievals depend on approximate temperature and pressure profiles that are computed via an assumed lapse rate, and this rate is computed from a surface observation (Spuler et al., 2015; Nehrir et al., 2009).

## 6.2 Data selection methodology

For the experiments we created datasets that span over specific months given 1) the high variability of WV across different
months and 2) the MPD WV measurement capability is dependent on the low altitude WV two-way transmittance. For all the datasets presented here, the observed photon counts are binned at range and time bins of 37.5 m by 5 minutes respectively. The





---

**Algorithm 2b** Coarse-to-fine image resolution framework for inferring WV $\varphi$ and atten. backs. $\chi$.

---

**Require:** 1) WV $\hat{\varphi}^{\text{init}}$ and atten. backs. $\hat{\tilde{\chi}}^{\text{init}}$ initial values, 2) WV $\lambda_w$ and atten. backs. $\lambda_a$ tuning parameters and training photon counts
$\mathbf{Y}_{\text{trn}}^{(\text{on})}$ and $\mathbf{Y}_{\text{trn}}^{(\text{off})}$, 3) coarsest resolution downsampling factor $\bar{h}$.

1:   $\varphi^{(\bar{h}+2)} = \hat{\varphi}^{\text{init}}, \tilde{\chi}^{(\bar{h}+2)} = \hat{\tilde{\chi}}^{\text{init}}$ {Set initial values}

2:   $h = \bar{h}$ {Iterate over the coarse-to-fine downsampling factors}

3:   **while** $h \geq 1$ **do**

4:      {Set initial values for current image resolution; downsample the initial WV estimate to the image resolution at which inference is done}

5:      $t = 0, \phi^{(t)} = \mathcal{D}_h^{\downarrow} \varphi^{(h+2)}, \psi^{(t)} = \tilde{\chi}^{(h+2)}$

6:      {Find minimizer of objective function (19) by alternating the minimization between the WV $\phi$ and atten. backs. $\psi$.}

7:      **repeat**

8:        {Solve for WV via adapted SPIRAL by minimizing Eq. (19); $\mathcal{W}$ is the set of non-negative numbers}

$$\phi^{(t+1)} = \arg \min_{\phi \in \mathcal{W}} F_{p_{\text{trn}}, \lambda_w, \lambda_a} \left( \mathcal{D}_h^{\uparrow} \phi, \psi^{(t)}; \right.$$
$$\left. \mathbf{Y}_{\text{trn}}^{(\text{on})}, \mathbf{Y}_{\text{trn}}^{(\text{off})} \right) \tag{28}$$

9:        {Solve for atten. backs. via adapted SPIRAL by minimizing Eq. (19)}

$$\psi^{(t+1)} = \arg \min_{\psi} F_{p_{\text{trn}}, \lambda_w, \lambda_a} \left( \mathcal{D}_h^{\uparrow} \phi^{(t+1)}, \psi; \right.$$
$$\left. \mathbf{Y}_{\text{trn}}^{(\text{on})}, \mathbf{Y}_{\text{trn}}^{(\text{off})} \right) \tag{29}$$

10:      $t = t + 1$

11:      **until** stopping criteria is Eq. (24) met

12:      {Set initial values for finer image resolution}

13:      $\varphi^{(h)} = \mathcal{D}_h^{\uparrow} \phi^{(t)}, \tilde{\chi}^{(h)} = \psi^{(t)}, h = \max(h - 2, 1)$

14: **end while**

15: {Set final estimates of WV and atten. backs.}

16: $\hat{\varphi}_{\bar{h}, \lambda_w, \lambda_a} = \varphi^{(1)}, \hat{\tilde{\chi}}_{\bar{h}, \lambda_w, \lambda_a} = \tilde{\chi}^{(1)}$

17: **return** $\hat{\varphi}_{\bar{h}, \lambda_w, \lambda_a}, \hat{\tilde{\chi}}_{\bar{h}, \lambda_w, \lambda_a}$

---

analyzed data from SGP is not comprehensive and instead we selected a variety of challenging cases in an effort to identify potential issues in the PTV-MPD algorithm. In particular much of the data targeted instances where clouds and precipitation created challenging scenes to processes.

Specific to the MPD at SGP the WV measurements start at 500 m range due to the poor geometric overlap below this range. The next generation MPD deployed at the Marshall field site employed optically combined narrow and wide field of view





telescopes to improve low altitude geometric overlap and as a result the WV measurements start at 150 m range (Spuler et al., 2021).

The first column of Table 4 lists the datasets, where the name of each dataset encodes the location with the year and month

interval that each dataset covers. The second column lists the specific dates that are included in each dataset. The third and fourth columns list the number of days in each dataset and the total number of coincident RS profiles.

### 6.3    PTV-MPD tuning parameters and coarse-to-fine (`PTV-CF`) configuration

The PTV-MPD method has the following input parameters as indicated in Alg. 2.

1.  The initial WV value $\hat{\varphi}^{\text{init}}$.

2.  The WV $\lambda_w$ and atten. backs. $\lambda_a$ tuning parameters that set the degree to which the 2D piecewise constant regularization is promoted for the WV $\hat{\varphi}$ and atten. backs. $\hat{\chi}$ geophysical variables; see (12).

3.  The coarsest resolution downsampling factor $\bar{h}$ which controls at what image resolution the WV inference starts at.

Also, the PTV-MPD method depends on an input cloud-mask which in itself is set by photon rate threshold (see Sect. 3.2). As discussed in Sect. 5.3.2 the initial WV value is set to zero $[\text{gm}^{-3}]$. The WV $\lambda_w$ and atten. backs. $\lambda_a$ tuning parameters are

determined through a cross-validation methodology (Friedman et al., 2001, chap. 7)(Oh et al., 2013); see Sect. 5.3.4 for more detail. For all the experiments PTV-MPD uses the default set of TV regularizer tuning parameters which are set between lines 1 to 5 in Alg. 2.

In regards to the resolution downsampling factor $\bar{h}$, we juxtapose the PTV-MPD method with ($\bar{h} > 1$) and without ($\bar{h} = 1$) the coarse-to-fine configuration to test the hypothesis whether the PTV-MPD coarse-to-fine image resolution framework does

improve the WV due to inaccuracies in the initial atten. backs. value. The test error (22) is used to quantify the hypotheses testing as indicated in (23). We use mnemonics, as indicated in Table 5, to make a distinction between the two PTV-MPD configurations and the standard method. The coarse-to-fine PTV-MPD method, `PTV-CF`, starts at a coarse image resolution which is nine times ($\bar{h} = 9$) coarser than the finest image resolution.

In the following subsection we show individual PTV-MPD WV measurement results and thereafter we present the WV

measurement error statistics.

### 6.4    Individual retrieval results

Figures 3 and 4 show the WV measurements obtained using the standard and PTV-MPD methods (see Table 5) when the MPD was at SGP for dates 11th and 10th June 2019. Fig. 5 show the WV measurements when the MPD was at Marshall on the 10th of December 2020. These examples have been selected to highlight instances of improvement as well as ongoing challenges

for the `PTV-CF` and `PTV` methods. For all these figures, sub-figure (a) shows the atten. backs. $\chi$. Sub-figures (b) to (d) show the WV $\varphi$ measurements of the `STND`, `PTV-CF` and `PTV` methods. The rest of the sub-figures, (e) and beyond, show the WV profiles of the different methods compared with that of the RS and Raman lidar; the white vertical dashed-lines in sub-figures

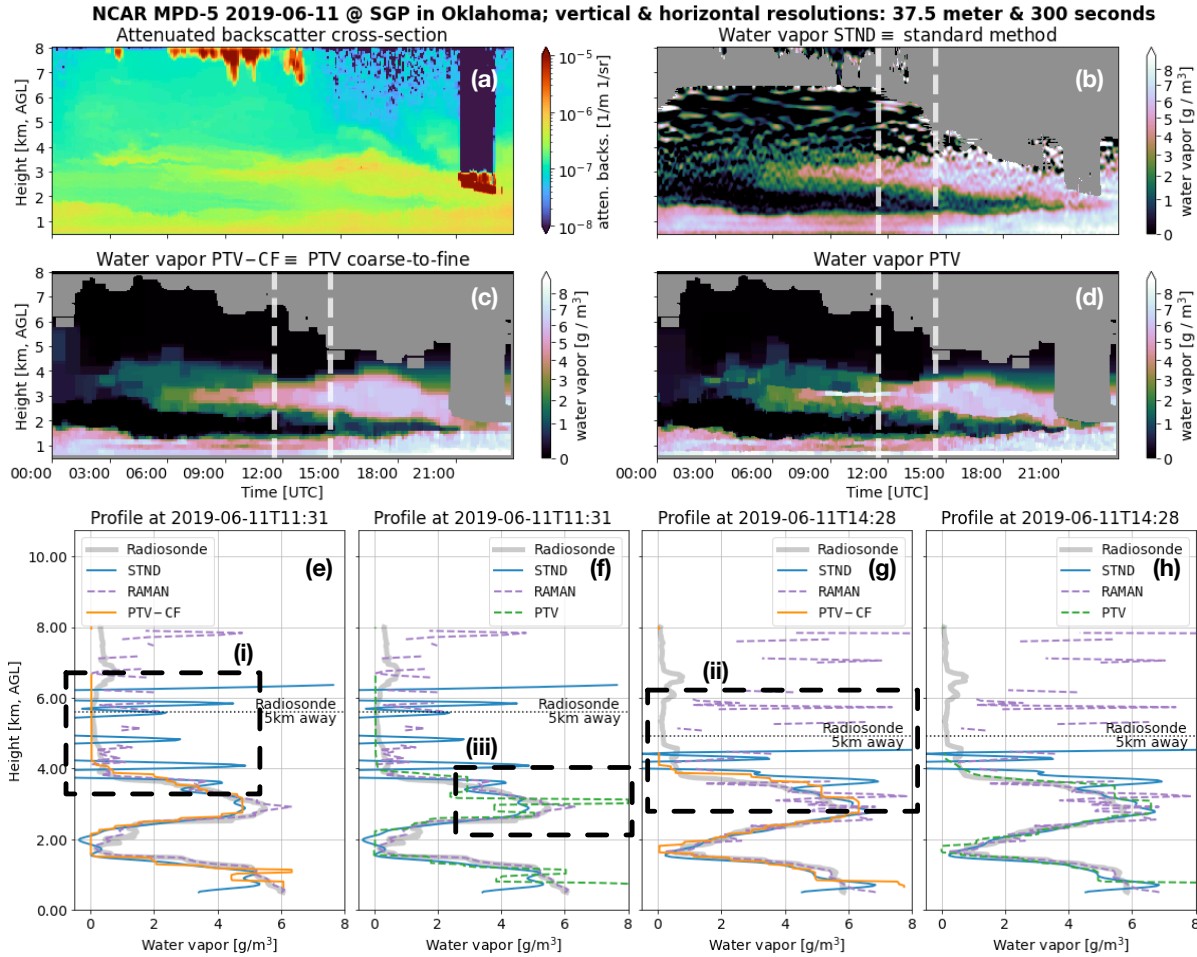

**Figure 3.** The (a) attenuated backscatter cross-section, the water vapor (WV) measurements of the (b) standard method (`STND`) and the PTV-MPD method (c) with (`PTV-CF`) and (d) without (`PTV`) the coarse-to-fine configuration. (a) is not masked to show what atmospheric features have been been masked out in (b) to (d) by the gray areas. The white dashed vertical lines in (b) to (d) show the launch times of the radiosondes (RSs), where (e) to (h) show the RS and Raman lidar WV profiles. (e) and (g) compares the course-to-fine PTV-MPD WV measurements against that of the RS and the standard method, whereas (f) and (h) show the same comparison for the PTV-MPD method without the course-to-fine configuration. The horizontal dashed lines in (e) to (h) show the altitude at which the RS was horizontally 5 km away from the MPD. The dashed boxes in (e) and (g) highlight regions that are discussed in the text.

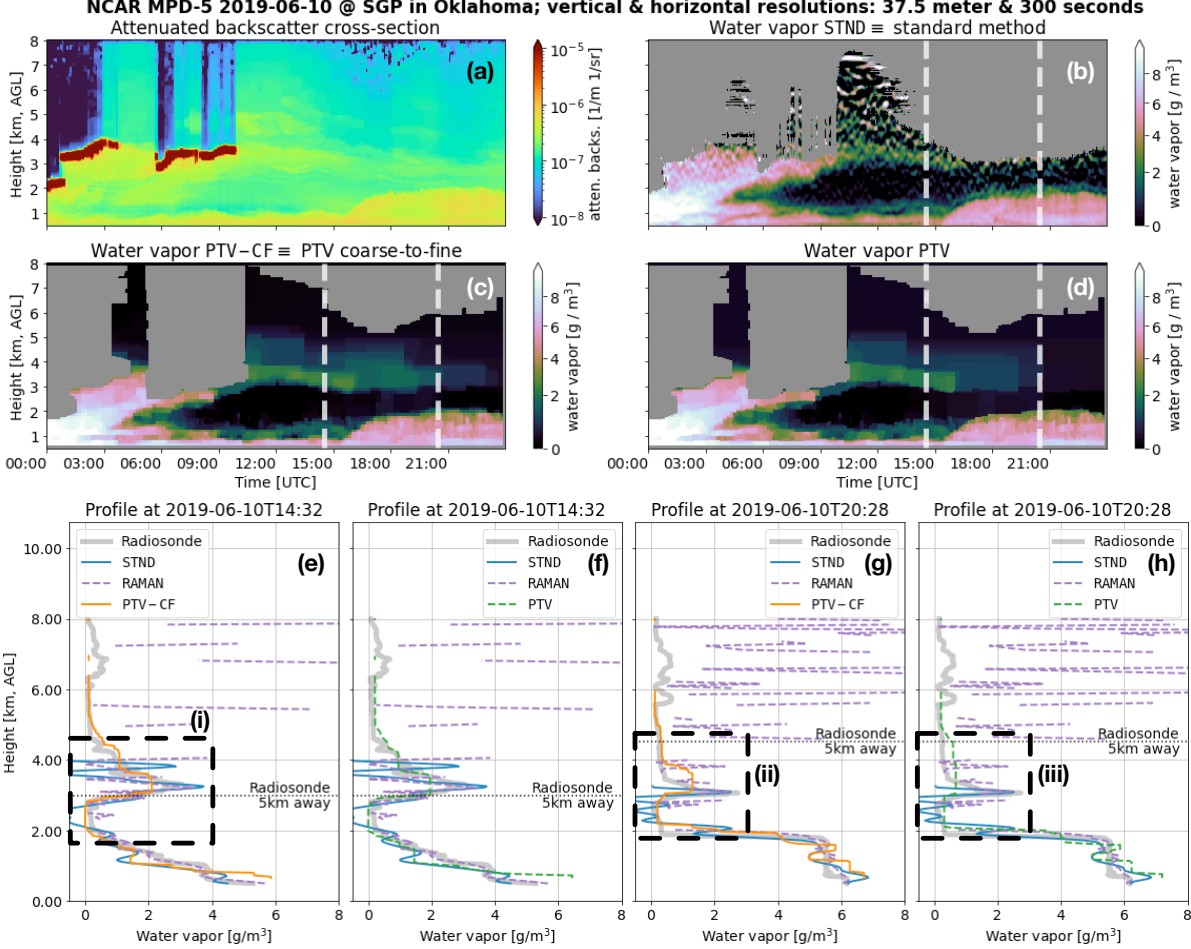

**Figure 4.** The (a) attenuated backscatter cross-section, the water vapor (WV) measurements of the (b) standard method (STND) and the PTV-MPD method (c) with (PTV−CF) and (d) without (PTV) the coarse-to-fine configuration. (a) is not masked to show what atmospheric features have been been masked out in (b) to (d) by the gray areas. The white dashed vertical lines in (b) to (d) show the launch times of the radiosondes (RSs), where (e) to (h) show the RS and Raman lidar WV profiles. (e) and (g) compares the course-to-fine PTV-MPD WV measurements against that of the RS and the standard method, whereas (f) and (h) show the same comparison for the PTV-MPD method without the course-to-fine configuration. The horizontal dashed lines in (e) to (h) show the altitude at which the RS was horizontally 5 km away from the MPD. The dashed boxes in (e) and (g) highlight regions that are discussed in the text.

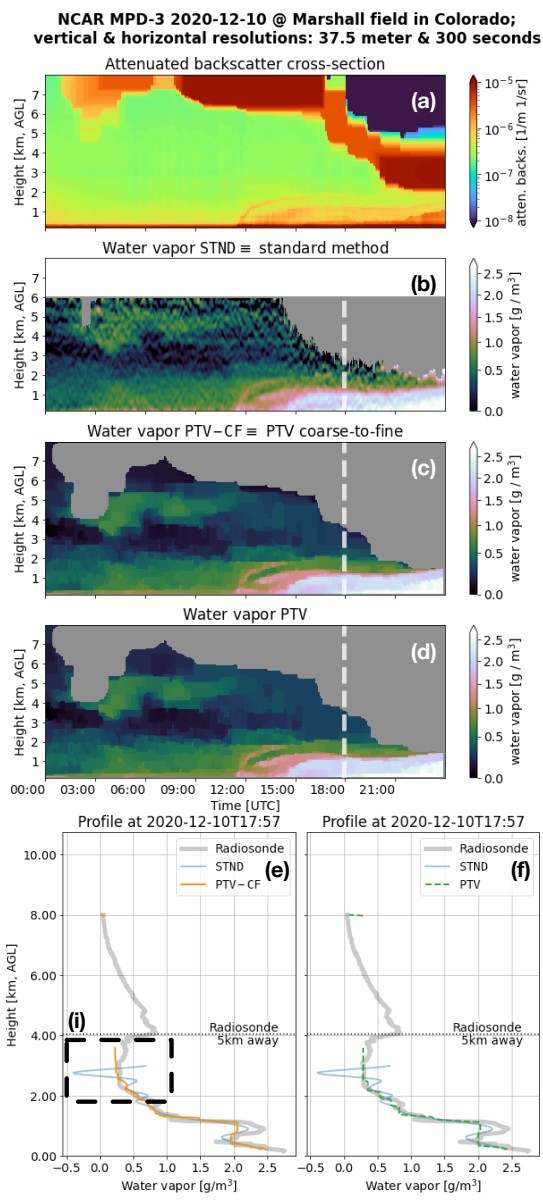

**Figure 5.** The (a) attenuated backscatter cross-section, the water vapor (WV) measurements of the (b) standard method (STND) and the PTV-MPD method (c) with (PTV-CF) and (d) without (PTV) the coarse-to-fine configuration. (a) is not masked to show what atmospheric features have been been masked out in (b) to (d) by the gray areas. The white dashed vertical lines in (b) to (d) show the launch times of the radiosondes (RSs), where (e) and (f) show the RS WV profiles. (e) compares the course-to-fine PTV-MPD WV measurements against that of the RS and the standard method, whereas (f) show the same comparison for the PTV-MPD method without the course-to-fine configuration. The horizontal dashed lines in (e) to (f) show the altitude at which the RS was horizontally 5 km away from the MPD. The dashed box in (e) highlights a region that is discussed in the text.



(b) to (d) show the launch times of the RSs. Sub-figure (a) is not masked to show what atmospheric features have been been masked out in sub-figures (b) to (d) by the gray areas.

### 6.4.1 Comparing `PTV` and `PTV-CF` with `STND`

When comparing the `PTV` and `PTV-CF` WV measurements with that of the `STND` method, the `STND` method exhibits higher residual noise particularly at higher altitudes as indicated by Figures 3.i, 3.ii, 4.i and 5.i. These oscillatory residual noise artifacts in the `STND` WV measurements are due to the low pass filter that is designed for high fidelity WV measurements up to 4 km. In contrast, the `PTV-CF` and `PTV` methods produce more accurate WV measurements at higher altitudes and these measurements do not exhibit large residual noise artifacts. The reasons why `PTV-CF` and `PTV` make more accurate high altitude WV measurements are because

1. `PTV-CF` and `PTV` employ the Poisson noise model with the MPD forward models to fit WV estimates onto the noisy observations and

2. via the TV regularization the `PTV-CF` and `PTV` methods exploit the spatial and temporal correlations of the WV to separate the WV from the random noise.

In some cases, residual noise in the `STND` method may falsely appear to capture RS observed WV structure. An example of this is shown in Fig. 4.ii where an elevated WV layer appears to be better captured by the `STND` method than `PTV-CF`, where `PTV-CF` identified an elevated layer that consists of lower density and greater vertical extent than the RS and Raman lidar. The uncertainty analysis, however, available with the standard method (i.e. `STND`) shows that the noise standard deviation in the WV estimate is greater than $2 \, [\mathrm{gm}^{-3}]$, indicating that the apparent structure is not statistically significant and is predominantly noise. This fact also becomes more apparent when viewing the time resolved plot in Fig. 4.b.

Fig. 4.ii demonstrates the challenge in validating the MPD profiles, particularly for regions exhibiting significant structure at higher altitudes. In this instance, the Raman lidar starts to struggle measuring WV at 3.5 km altitude during the daytime, due to the high solar background radiation noise. In order to validate such cases the MPD, Raman lidar and RS instruments should ideally make WV measurements of the same atmospheric volume which is not always possible. For the purposes of the statistical analysis presented in the next section, we will nevertheless treat the RS observations as "truth" despite possible discrepancies resulting from difference in atmospheric volume.

### 6.4.2 Comparing `PTV-CF` with `PTV`

Comparing Figure 3.c and 3.d we see that the `PTV` WV measurement appears to be larger than that of `PTV-CF` between 9 and 12 UTC at 3km. Fig. 3.iii quantitatively shows the large `PTV` WV measurement which appears to be an artifact. In contrast, the corresponding `PTV-CF` WV measurement in Fig. 3.e is closer to RS WV. The likely cause of the artifact produced by `PTV` is due to inaccuracies in the initial atten. backs. which are induced by

1. the assumed WV value when computing the initial value of the atten. backs. (see Sect. 5.3.2) and



2. long laser pulse, denoted by operator $\mathbf{A}$ in Eq. (5), that is not deconvolved from initial atten. backs.

Figures 4.ii and 4.iii show another distinct WV measurement difference between the `PTV-CF` and `PTV` methods. The tenuous
WV identified by `PTV-CF` correlates more with the RS WV measurement in magnitude and shape compared to the `PTV`
method.

The higher accuracy of the `PTV-CF` WV measurements corresponding to Figures 3.iii and 4.iii is because the initial values of
the atten. backs. and WV are systematically improved with the coarse-to-fine image resolution framework. Specifically, while
inferring the atten. backs. and WV from both the on- and offline channels simultaneously at the coarsest image resolution, the
`PTV-CF` method:

1. implicitly deconvolves the long laser pulse from the initial atten. backs.,

2. and inference of the WV is constrained to be at a coarse image resolution which implicitly increases the SNR of the
   observations.

## 6.5 Water vapor measurement error statistics

The RS WV measurements are used as reference to produce WV measurement error statistics of the different methods which
is discussed in the following subsection. Next we discuss the test errors between the `PTV-CF` and `PTV` methods. We then
summarize the results of the error statistics.

### 6.5.1 Radiosonde comparisons

For each dataset we compute the root mean squared "error" (RMSE) per range and the relative RMSE per range; all the methods
use the same mask to exclude the same WV pixels when computing the RMSE. The RMSE and relative RMSE were computed
as follows. Let $\hat{\varphi}_n^{(1)}, \hat{\varphi}_n^{(2)}, \ldots, \hat{\varphi}_n^{(L)}$ denote the WV profiles inferred by one of the methods, and let $\varphi_n^{(1)}, \varphi_n^{(2)}, \ldots, \varphi_n^{(L)}$ denote
coincident RS WV profiles. The common mask profiles are denoted by $\widetilde{\mathbf{M}}_n^{(1)}, \widetilde{\mathbf{M}}_n^{(2)}, \ldots, \widetilde{\mathbf{M}}_n^{(L)}$; for example $\widetilde{\mathbf{M}}_n^{(1)} = 0$ indicates
that the WV measurement at row index $n$ is masked out by either the standard or PTV-MPD method. The RMSE and relative
RMSE (RRMSE) per dataset are computed by

$$N^{(l)} = \sum_{n=1}^{N} \widetilde{\mathbf{M}}_n^{(l)} \tag{30}$$

$$\mathrm{RMSE} = \left( \frac{1}{L} \sum_{l=1,n=1}^{L,N} \frac{\widetilde{\mathbf{M}}_n^{(l)}}{N^{(l)}} \left( \hat{\varphi}_n^{(l)} - \varphi_n^{(l)} \right)^2 \right)^{1/2}, \tag{31}$$

$$\mathrm{RRMSE} = 100 \times \mathrm{RMSE}$$
$$\times \left( \frac{1}{L} \sum_{l=1,n=1}^{L,N} \frac{\widetilde{\mathbf{M}}_n^{(l)}}{N^{(l)}} \left( \varphi_n^{(l)} \right)^2 \right)^{-1/2}. \tag{32}$$





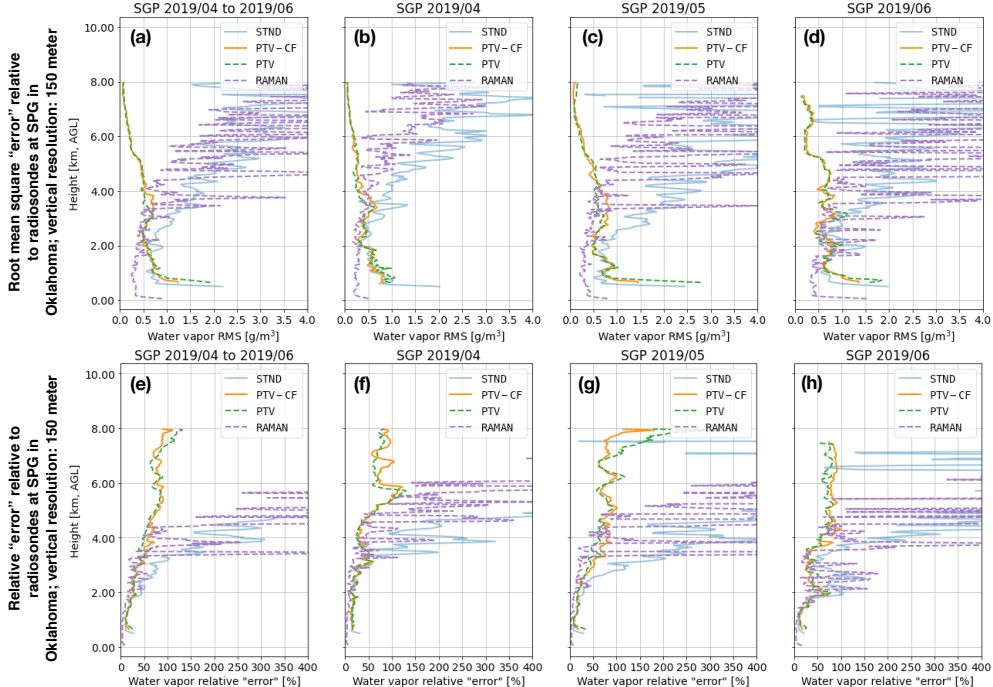

**Figure 6.** For the all the SGP datasets (a) show the root mean squared "error" (RMSE) relative to the radiosonde (RS) water vapor (WV) measurements for the standard method (STND) and the PTV-MPD method with (PTV-CF) and without (PTV) the coarse-to-fine configuration; (e) show the corresponding relative RMSE. (b) to (d) show the RMSE for each individual SGP dataset, and (f) to (h) show the corresponding relative RMSE.

Figures 6.a and 7.a show the RMSE over the whole SG and Marshall datasets, respectively, whereas Figures 6.e and 7.e show the corresponding RRMSE. Sub-figures (b) to (d) of Figures 6 and 7 show the RMSE for each SGP and Marshall dataset (see Table 4), whereas sub-figures (f) to (h) show the corresponding RRMSE.

     From Figures 6 and 7 we see that above 1.5 km the PTV-CF and PTV RMSEs and RRMSEs are comparable. Except for the lowest altitudes in Figures 6.c and 6.d the PTV-CF RMSEs are approximately less than 1 $[\mathrm{gm}^{-3}]$ for all altitudes whereas
the STND RMSE degrades with increasing altitude. For the SGP datasets the PTV-CF and PTV RRMSEs are less than 100% in most of the profiles up to 8 km altitude, where for the SGP 2019/04 dataset in Figure 6.f the RRMSE peaks at 125 % near 6 km altitude. The PTV-CF and PTV RRMSEs for the Marshall datasets in Fig. 7 are similar to the SGP RRMSEs below 8 km altitude.

     The Ramar lidar outperforms the PTV-CF and PTV when its observations have sufficient high SNR. Otherwise in high
solar background conditions (i.e. lower SNR) the Raman lidar's effective range is diminished. By employing PTV-CF or PTV, the MPD, which had approximately 500 times lower power-aperture-product than the Raman lidar, is able to obtain higher accuracy WV measurements above 4 km compared to the Raman lidar.



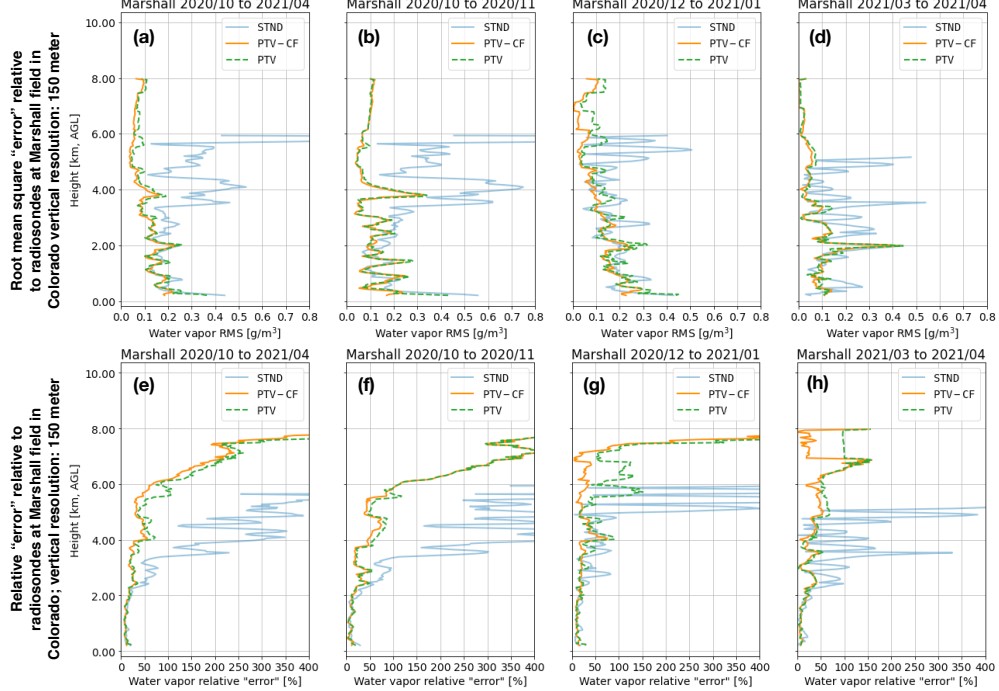

**Figure 7.** For the all the Marshall datasets (a) show the root mean squared "error" (RMSE) relative to the radiosonde (RS) water vapor (WV) measurements for the standard method (`STND`) and the PTV-MPD method with (`PTV-CF`) and without (`PTV`) the coarse-to-fine configuration; (e) show the corresponding relative RMSE. (b) to (d) show the RMSE for each individual Marshall dataset, and (f) to (h) show the corresponding relative RMSE. Note that the scale of the horizontal axis in (a) to (d) is smaller than that shown in Figure 6.

From Figures 6.a and 7.a we see that the lowest altitude WV measurements of the `PTV-CF` method is more accurate on average compared to the `PTV` and `STND` methods. These lower WV accuracies of the `PTV` and `STND` methods are due to
lower SNR near range observations and inaccuracies in the initial atten. backs. As explained in Sect. 6.4.2, the course-to-fine methodology employed by `PTV-CF` mitigates the inaccuracies of the initial atten. backs.

### 6.5.2 Test error comparisons

Table 6 shows that for five out of the six datasets the `PTV-CF` test errors ($\mathrm{err}_{\mathrm{tst}, \bar{h}=9}$) are less than the `PTV` test error ($\mathrm{err}_{\mathrm{tst}, \bar{h}=1}$). The lower test errors of `PTV-CF` correlates with the higher WV accuracy that `PTV-CF` is able to achieve compared to `PTV` at
the lower altitudes.

### 7 Conclusion and future work

PTV-MPD employs the photon counting noise model to quantitatively fit the MPD forward models on the noisy observations, while encouraging accurate spatial and temporal correlations across all WV estimate pixels via the total variation regularizer



function. This holistic approach allows for accurately measuring highly structured and varying WV fields at different altitudes
and varying SNRs. In comparison, the standard method employs low pass filters to reduce the residual noise in the WV
estimate and the low pass filter bandwidth is optimized for lower altitude ($\leq 4$ km) WV measurements. Therefore, the standard
method high altitude WV measurements contains residual noise that degrades the measurements. By applying PTV-MPD to
WV retrievals from the MPD, we have been able to extend the maximum altitude of the retrieval from 6 km to 8 km, enabling a
maximum coverage (after hardware improvements described in (Spuler et al., 2021)) from 150 m to 8 km by the instrument. In
addition, the WV retrieval accuracy is substantially increased above 2 km. It is also notable that by employing the PTV-MPD
method, the MPD WV measurement range extends beyond the that of the ARM-SGP Raman lidar which has nearly 500 times
the power-aperture-product of the MPD (Newsom and Sivaraman, 2018).

We also demonstrated that without careful consideration of how PTV is adapted for the MPD instrument, low altitude
biases can be introduced in the PTV-MPD WV measurements. PTV-MPD requires an initial value of the atten. backs. and any
inaccuracies can induce biases in the PTV-MPD WV estimates. PTV-MPD can be made more robust against such biases by
inferring the WV via a coarse-to-fine image resolution framework; by inferring the atten. backs. and WV at a coarse image
resolution inaccuracies in the initial atten. backs. are reduced, and subsequent finer image resolution atten. backs. estimates are
more accurate which allows for more accurate WV estimates.

As of now PTV-MPD is computationally expensive since inferring the WV requires estimating the WV with several tuning
parameters, and the optimal tuning parameter is selected through a cross-validation methodology. For example, with 144 CPU
cores it takes 1 to 2 hours to infer the WV using the PTV-MPD coarse-to-fine image resolution framework for 24 hours of data;
each CPU core estimates the WV for a specific tuning parameter. We are working towards developing a methodology to infer
the optimal tuning parameter from the photon counting observations which will reduce the number of required CPU cores to
12 or less. In addition, adapting the PTV-MPD code to use a GPU instead of a CPU might reduce the computational time by at
least 10-fold (Lee and Wright, 2008).

An additional benefit of decreasing the computational demands of PTV-MPD would be able to process multiple days of
consecutive observations and not just 24-hour scenes as demonstrated in the results. A current workaround to process multiple
day scenes is to use a horizontal sliding window when inferring the WV. Specifically, the WV is inferred for consecutive
overlapping 24-hour periods, and the final multiple day WV image is obtained by averaging together the overlapping 24-hour
period WV estimates.

Future work includes:

1. Working towards quantifying the uncertainties of the PTV-MPD WV measurements using a bootstrapping methodol-
   ogy (Friedman et al., 2001). The uncertainty quantification will quantify how PTV-MPD WV measurements behave at
   the edges of scenes where less spatial information is available.

2. Investigating how PTV-MPD can be made more robust against saturated photon counts.





*Code availability.* PTV-MPD code will be made available through the GIT repository https://github.com/WillemMarais/poisson-total-variation in the near future.

*Data availability.* All the MPD data products used in this work were produced by NCAR and are available upon request. The SGP radiosonde data are available via https://adc.arm.gov/discovery (last access: 19 October 2021, (ARM, 2021)). The Marshall radiosonde data are available
570   via the GCOS Reference Upper-Air Network (GRUAN) website https://www.gruan.org (last access: 19 October 2021).

**Appendix A: Jacobian matrix of MPD forward models Eq. (5) and gradient matrix of loss function Eq. (12)**

Define the indicator function as

$$
\mathbb{I}\{k = l\} =
\begin{cases}
1 & \text{if } k = l \\
0 & \text{if } k \neq l.
\end{cases}
\tag{A1}
$$

The MPD forward model derivatives relative to WV and atten. backs. are

$$
\quad \frac{\partial}{\partial \boldsymbol{\varphi}_{j,l}} \mathbf{S}_{n,k}^{(\iota)}(\boldsymbol{\varphi}, \exp(\tilde{\boldsymbol{\chi}})) = -2\Delta r \Delta t \boldsymbol{\sigma}_{j,l}^{(\iota)} \boldsymbol{U}_k \mathbb{I}\{k = l\}
\tag{A2}
$$

$$
\times \sum_{n'=n}^{n+\Delta N} \boldsymbol{A}_{n',n}^T \mathbb{I}\{j \leq n'\} \tilde{\boldsymbol{S}}_{n',k}^{(\iota)}(\boldsymbol{\varphi}, \exp(\tilde{\boldsymbol{\chi}})),
\tag{A3}
$$

$$
\frac{\partial}{\partial \tilde{\boldsymbol{\chi}}_{j,l}} \mathbf{S}_{n,k}^{(\iota)}(\boldsymbol{\varphi}, \exp(\tilde{\boldsymbol{\chi}})) = \Delta t \boldsymbol{U}_k \mathbb{I}\{k = l\}
\tag{A4}
$$

$$
\times \sum_{n'=n}^{n+\Delta N} \boldsymbol{A}_{n',n}^T \mathbb{I}\{j \leq n'\} \tilde{\boldsymbol{S}}_{n',k}^{(\iota)}(\boldsymbol{\varphi}, \exp(\tilde{\boldsymbol{\chi}}))
\tag{A5}
$$

The P-NLL loss function derivatives relative to WV and atten. backs. are

$$
\quad \frac{\partial}{\partial \boldsymbol{\varphi}_{j,l}} \mathcal{L}_p \left( \mathbf{S}^{(\iota)}(\boldsymbol{\varphi}, \exp(\tilde{\boldsymbol{\chi}})); \mathbf{Y}^{(\iota)} \right) = -2\Delta r \Delta t p \boldsymbol{\sigma}_{j,l}^{(\iota)}
\tag{A6}
$$

$$
\times \boldsymbol{U}_l \sum_{n'=j}^{N+\Delta N} \tilde{\boldsymbol{S}}_{n',l}^{(\iota)}(\boldsymbol{\varphi}, \exp(\tilde{\boldsymbol{\chi}})) \sum_{n=\max(n'-\Delta N,1)}^{\min(n',N)} \boldsymbol{A}_{n',n}^T
\tag{A7}
$$

$$
\times \boldsymbol{M}_{n,l} \left( 1 - \frac{\boldsymbol{Y}_{n,l}^{(\iota)}}{p\boldsymbol{S}_{n,l}^{(\iota)}(\boldsymbol{\varphi}, \exp(\tilde{\boldsymbol{\chi}}))} \right),
\tag{A8}
$$

$$
\frac{\partial}{\partial \tilde{\boldsymbol{\chi}}_{j,l}} \mathcal{L}_p \left( \mathbf{S}^{(\iota)}(\boldsymbol{\varphi}, \exp(\tilde{\boldsymbol{\chi}})); \mathbf{Y}^{(\iota)} \right) = \Delta t p
\tag{A9}
$$

$$
\times \boldsymbol{U}_l \tilde{\boldsymbol{S}}_{j,l}^{(\iota)}(\boldsymbol{\varphi}, \exp(\tilde{\boldsymbol{\chi}})) \sum_{n=\max(n'-\Delta N,1)}^{\min(n',N)} \boldsymbol{A}_{n',n}^T
\tag{A10}
$$

$$
\quad \times \boldsymbol{M}_{n,l} \left( 1 - \frac{\boldsymbol{Y}_{n,l}^{(\iota)}}{p\boldsymbol{S}_{n,l}^{(\iota)}(\boldsymbol{\varphi}, \exp(\tilde{\boldsymbol{\chi}}))} \right).
\tag{A11}
$$



**Appendix B: Poisson negative log likelihood Eq. (8) properties**

Here we show that for high solar background radiation (i.e. lower SNR) of the photon counting images there can be multiple estimates of the WV $\varphi$ or atten. backs. $\tilde{\chi}$ that minimizes the objective function (14). Specifically, we show that the P-NLL (8) has more the one minimizer. Without loss of generality we assume here that

1. the laser pulse duration is equal to the sampling intervals meaning that $\Delta N = 0$ and the matrix $\mathbf{A}$ is just an identity matrix,

    2. none of the photon counts are masked out by matrix $\mathbf{M}$,

    3. and $p = 1$.

With these assumptions the Hessian matrix for observation column $k$ of the P-NLL (8) relative to WV $\varphi$ for given atten. backs.

$\chi$ equals

$$\boldsymbol{H}_{j,l}(\boldsymbol{\varphi};\boldsymbol{\chi}) = 4\Delta r^2 \boldsymbol{\sigma}_{j,k}^{(\iota)} \boldsymbol{\sigma}_{l,k}^{(\iota)} \sum_{l'=l}^{N} \mathbb{I}\{j \le l'\} \tag{B1}$$

$$\times \left( \tilde{\boldsymbol{S}}_{l',k}^{(\iota)}(\boldsymbol{\varphi},\boldsymbol{\chi}) - \boldsymbol{U}_k \Delta t \boldsymbol{b}_k \right) \tag{B2}$$

$$\times \left( 1 - \boldsymbol{Y}_{l',k}^{(\iota)} \frac{\boldsymbol{U}_k \Delta t \boldsymbol{b}_k}{\left[ \tilde{\boldsymbol{S}}_{l',k}^{(\iota)}(\boldsymbol{\varphi},\boldsymbol{\chi}) \right]^2} \right). \tag{B3}$$

If this Hessian matrix is positive definite, meaning all its eigenvalues positive, then it implies that for a fixed atten. backs. $\chi$

there is a unique WV $\varphi$ estimate that minimizes the P-NLL and therefore the objective function (14) (Boyd and Vandenberghe, 2004). The Hessian matrix is not positive definite if

$$\tilde{\boldsymbol{S}}_{l',k}^{(\iota)}(\boldsymbol{\varphi},\boldsymbol{\chi}) \le \left[ \boldsymbol{Y}_{l',k}^{(\iota)} \boldsymbol{U}_k \Delta t \boldsymbol{b}_k \right]^{\frac{1}{2}}. \tag{B4}$$

Whether this inequality is met is dependent on the atten. backs., WV transmittance and laser energy. For example, for humid summer days the WV transmittance will significantly decrease the SSLE relative to the solar background radiation, conse-

quently limiting PTV-MPD's ability to uniquely infer the WV at higher altitudes.

    In the cases where the backscattered photons are comparable to the observed background counts according to (B4), we leave it to future work to determine the altitudes at which PTV-MPD can uniquely infer the WV.

*Author contributions.* This work was a collaboration between the authors. Marais is the original developer of PTV and provided expertise in advanced signal processing. Hayman provided expertise in optical system modeling and DIAL observation. The PTV-MPD development

was the result of a collaborative effort to mutually leverage the two disciplines.





*Competing interests.* The authors declare that they have no conflict of interest.

*Financial support.* This material is based upon work supported by NSF AGS-1930907 and the National Center for Atmospheric Research, which is a major facility sponsored by the National Science Foundation under Cooperative Agreement No. 1852977.

*Acknowledgements.* The authors would like to thank Robert A. Stillwell and Scott M. Spuler for their feedback on this paper.



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





| Symbol | Type | Description |
|---|---|---|
| $n, n', n''$ | Index | Matrix row index |
| $k, k', k''$ | Index | Matrix column index |
| $\iota$ | Index | On- and offline channel index where $\iota \in \{\text{on}, \text{off}\}$ |
| $\Delta N$ | Scalar | The number of vertical sampling intervals that fit in the length of the laser pulse duration, minus one |
| $\boldsymbol{r}_{n'}$ | Vector | Range axis of the single scatter lidar equation and the geophysical variables |
| $\tilde{\boldsymbol{r}}_n$ | Vector | Range axis of the observations as modeled by the MPD forward models; $\tilde{\boldsymbol{r}}_n = \boldsymbol{r}_{n+\lceil \Delta N/2 \rceil}$ |
| $\boldsymbol{\chi}_{n',k}$ | Matrix | Uncalibrated attenuated backscatter [m$^{-1}$sr$^{-1}$] of size $(N + \Delta N) \times K$ |
| $\boldsymbol{\varphi}_{n',k}$ | Matrix | Absolute water vapor [gm$^{-3}$] of size $(N + \Delta N) \times K$ |
| $\tilde{\mathbf{S}}_{n',k}^{(\iota)}(\boldsymbol{\chi}, \boldsymbol{\varphi})$ | Matrix | MPD channel $\iota$ single scatter lidar equation matrix function [W] of size $(N + \Delta N) \times K$ |
| $\mathbf{A}_{n,n'}$ | Matrix | Model for laser energy distribution over sampling intervals |
| $\mathbf{S}_{n,k}^{(\iota)}(\boldsymbol{\chi}, \boldsymbol{\varphi})$ | Matrix | MPD channel $\iota$ forward model matrix function [J] of size $N \times K$ |
| $\mathbf{Y}_{n,k}^{(\iota)}$ | Matrix | Accumulated photon count of channel $\iota$ where its expected value is modeled by $\mathbf{S}_{n,k}^{(\iota)}(\boldsymbol{\chi}, \boldsymbol{\varphi})$ |
| $\boldsymbol{b}_k^{(\iota)}$ | Vector | Solar and dark background rate [W] of size $K$ |
| $\mathbf{M}_{n,k}$ | Matrix | A binary mask value for accumulated photon count $\mathbf{Y}_{n,k}^{(\iota)}$ where a value of zero indicates that the photon count is saturation contaminated |
| $\boldsymbol{U}_k$ | Vector | The number of laser shots per column index $k$ |

Table 1. Commonly used symbols.





| Acronym | Elaboration |
|---|---|
| atten. backs. | Attenuated backscatter [$m^{-1}\ sr^{-1}$] |
| CV | Cross-validation |
| MPD | MicroPulse Differential absorption lidar |
| P-NLL | Poisson Negative Log Likelihood |
| PTV | Poisson Total Variation |
| RS | Radiosonde |
| RMSE | Root Mean Square Error |
| RRMSE | Relative RMSE |
| SNR | Signal-to-noise ratio |
| SSLE | Single scatter lidar equation [W] |
| TV | Total Variation |
| WV | Water vapor [$gm^{-3}$] |

**Table 2.** Commonly used acronyms.

| Lidar (location) | Laser power | Telescope diameter |
|---|---|---|
| Raman (SGP, OK) | 9 W | 61 cm |
| MPD (SGP, OK) | 45 mW | 40.6 cm |
| MPD (Marshall, CO) | 24 mW | 40.6 cm |

**Table 3.** The first column indicates the lidar instrument and where the instrument was located; SGP refers to the Southern Great Plains in Oklahoma and Marshall refers to Marshall field in Boulder Colorado. The second and third column show the corresponding laser power and telescope diameter.

| Dataset (location, year/month) | Months and days (month/day) | Number of days | Number of radiosonde profiles |
|---|---|---|---|
| SGP 2019/04 | 4/19, 4/20, 4/25, 4/27, 4/28 | 5 | 32 |
| SGP 2019/05 | 5/10, 5/12, 5/13, 5/14, 5/15, 5/16, 5/19 | 7 | 52 |
| SGP 2019/06 | 6/10, 6/11, 6/24, 6/29 | 4 | 31 |
| Marshall 2020/10 to 2020/11 | 10/16, 11/6, 11/12, 11/20, 11/25 | 5 | 5 |
| Marshall 2020/12 to 2021/01 | 12/10, 12/21, 12/30, 1/4, 1/13, 1/22, 1/29 | 7 | 7 |
| Marshall 2021/03 to 2021/04 | 3/1, 3/31, 4/2 | 3 | 3 |

**Table 4.** The first column shows the names of the datasets used in the experiments. The second column lists the specific dates that is included in each dataset. The third and fourth columns list the number of days in each dataset and the total number of coincident RS profiles.

| Algorithm | Alg. 1 | Alg. 2 with $\bar{h} = 9$ | Alg. 2 with $\bar{h} = 1$ |
|---|---|---|---|
| Mnemonic | STND | PTV-CF | PTV |

**Table 5.** The mnemonics used to indicate the algorithm that was used to produce a water vapor estimate, and the corresponding test errors.





| Dataset | $\mathrm{err}_{\mathrm{tst},\bar{h}=9} < \mathrm{err}_{\mathrm{tst},\bar{h}=1}$ |
|---|---|
| SGP 2019/04 to 2019/06 | True |
| Marshall 2020/10 to 2021/04 | True |
| SGP 2019/04 | True |
| SGP 2019/05 | True |
| SGP 2019/06 | True |
| Marshall 2020/10 to 2020/11 | False |
| Marshall 2020/12 to 2021/01 | True |
| Marshall 2021/03 to 2021/04 | True |

**Table 6.** This table shows for which datasets the test error of the PTV-MPD method with the coarse-to-fine configuration ($\mathrm{err}_{\mathrm{tst},\bar{h}=9}$) is smaller (i.e. better) than PTV-MPD without the coarse-to-fine configuration ($\mathrm{err}_{\mathrm{tst},\bar{h}=1}$). The first two rows of the table, before the horizontal divider line, correspond to the total test error comparison over all the SGP and Marshall datasets. The rest of the rows corresponding to the test error comparison over the specific SGP and Marshall datasets.