# Peer review of "Extending water vapor measurement capability of photon limited differential absorption lidars through simultaneous denoising and inversion"

_Atmospheric Measurement Techniques, 2021_

## Author Response (AR1)

**Response to reviewers of amt-2021-352**

**Anonymous referee #2**

We thank the reviewer for the time spent on reading the paper in detail and providing us with helpful feedback. Here are the response to some of the specific comments.

**Statement:** This work is fairly specialized, being suited to photon-counting DIAL systems with pulsed lasers, and the MPD system in particular, so it may not find a wide audience.

**Response:** We agree with the referee except that PTV was also adapted for the photon counting high spectral resolution lidar (HSRL) instrument (Marais et al. 2016). This work demonstrates that PTV extends to applications beyond backscatter and HSRL systems and it is the first time we have been able to validate PTV retrievals against a gold standard (radiosondes). In that sense this work is a data point supporting the claim that PTV has broad application to improving data products from lidar remote sensors.

**Statement:** The authors go to some trouble to account for convolution with the relatively long laser pulses, (1 microsecond, ~300 m), compared to the 37.5 m vertical resolution displayed in the plots. Shorter laser pulses (~ns) can be easily achieved, even in low-cost laser diodes. What are the tradeoffs for using longer pulses? I imagine the longer pulses allow for more backscattered photons, requiring shorter acquisition times to achieve the same SNR, but at the expense of decreased vertical resolution. A detailed study of this is probably outside the scope of this paper, but a brief discussion of the qualitative trade offs would be nice.

**Response:** We briefly discuss the tradeoff of the long laser pulses in the introduction in section 1.2; here is the text we added (in italics):

"Forward models for DIAL: The first adaptation of PTV requires that we develop MPD forward models to fit the estimated parameters on to the observed photon counts; while the objective is to estimate the WV, to accommodate the forward model, the attenuated backscatter (atten. backs.) is also an estimated product. *To accurately model what the MPD observes a convolution operator is included in the forward model that represents the oversampling of the laser pulse (Spuler et al., 2021); the MPD uses relatively long laser pulses to increase the SNR with the trade off of blurring the backscatter optical signal described by the standard single scatter lidar equation (SSLE)."*

**Statement:** There are several minor grammatical or typographical errors, but overall the manuscript is very well written. A few specific examples with line number and suggested text:

**Response:** Thank you for noticing and pointing out these grammatical or typographical errors; all the suggestions were incorporated in the paper.

**Anonymous referee #3**

We thank the reviewer for the time spent on reading the paper in detail and providing us with helpful feedback. Here are the response to some of the specific comments.

**Statement:** Links are provided for the data sources. The code will be made accessible via github, but maybe a repository like the CERN repository zenodo.org, which offers a github integration would be a better choice as it allows to use DOI also for code repositories.

**Response:** Thank you for this suggestion, the authors will explore this repository option in the future. Currently we have  DOI for the raw MPD data (https://doi.org/10.26023/MX0D-Z722-M406).  For now, the PTV processed MPD data is available on request and the processed standard data is in the ARM data archive. Additionally future processed field data will be archived by NCAR's existing data management system.